# Mitigating the Likelihood Paradox in Flow-based OOD Detection via Entropy Manipulation

## Abstract

Deep generative models that can tractably compute input likelihoods, including normalizing flows, often assign unexpectedly high likelihoods to out-of-distribution (OOD) inputs. We mitigate this likelihood paradox by manipulating input entropy based on semantic similarity, applying stronger perturbations to inputs that are less similar to an in-distribution memory bank. We provide a theoretical analysis showing that entropy control increases the expected log-likelihood gap between in-distribution and OOD samples in favor of the in-distribution, and we explain why the procedure works without any additional training of the density model. We then evaluate our method against likelihood-based OOD detectors on standard benchmarks and find consistent AUROC improvements over baselines, supporting our explanation.

## 1 Introduction

Out-of-distribution (OOD) detection is important in fields such as manufacturing systems and medicine (Mezher & Marble, 2024; Narayanaswamy et al., 2023). A supervised, discriminative approach is possible, but it is often impractical because it assumes access to OOD data during training and sufficient coverage of diverse OOD types (Havtorn et al., 2021). To avoid this requirement, unsupervised methods based on deep generative models have been studied for OOD detection (Yu et al., 2021; Ryu et al., 2018; Ran et al., 2022). Three representative deep generative model families for this task are the variational autoencoder (VAE, Kingma & Welling (2014)), the generative adversarial network (GAN, Goodfellow et al. (2014)), and normalizing flows (Dinh et al., 2015). Among these models, normalizing flows are attractive because they provide tractable likelihood estimates, whereas VAEs typically provide only a lower bound on the likelihood and GANs do not compute tractable likelihoods (Nalisnick et al., 2019a). Since flows yield tractable likelihoods, one can score test inputs by their log-likelihood; under the usual assumption that in-distribution (ID) samples have higher likelihood, OOD samples should have lower log-likelihoods than ID samples.

However, several studies have reported **counterintuitive behavior in OOD detection** when likelihoods from normalizing flows are used as the detection statistic (Nalisnick et al., 2019a). For example, a model trained on CIFAR-10 (Krizhevsky et al., 2009) may assign higher likelihood to SVHN (Netzer et al., 2011) as an OOD dataset than to CIFAR-10 itself (Nalisnick et al., 2019a). To better understand this failure of likelihood-based OOD detection from a theoretical perspective, Caterini & Loaiza-Ganem (2022) decomposed the expected log-likelihood difference for ID $P$, OOD $Q$, and a model $P_\theta$ as[1]:

$$
\begin{aligned}
&\mathbb{E}_{\mathbf{x} \sim P}[\log P_\theta(\mathbf{x})] - \mathbb{E}_{\mathbf{x} \sim Q}[\log P_\theta(\mathbf{x})] \\
&= D_{KL}(Q\|P_\theta) - D_{KL}(P\|P_\theta) + \mathbb{H}(Q) - \mathbb{H}(P)
\end{aligned}
\tag{1}
$$

When $\mathbb{H}(P) > \mathbb{H}(Q)$, this difference can flip sign even if $P_\theta$ fits $P$ well (i.e., $D_{\mathrm{KL}}(P\|P_\theta)$ is small), which explains failures of raw likelihood as an OOD detection statistic. Motivated by the role of entropy in this

---

[1]Without loss of generality, we assume that $D_{KL}(Q\|P_\theta), D_{KL}(P\|P_\theta)$ exist.

decomposition, we manipulate test-time entropy so that likelihood-based OOD scores better reflect semantic typicality. Specifically, we use a pretrained feature extractor to encode semantic information. Our method perturbs inputs with Gaussian noise, where the noise scale depends on the maximum cosine similarity between the test embedding and a memory bank of ID embeddings. Notably, the procedure is post hoc and requires no additional training of the density model and similarity is used only as a control signal to manipulate entropy, while the final OOD score remains the model likelihood. We show both theoretically and empirically that semantically controlled entropy-increasing perturbations increase the expected log-likelihood gap. Furthermore, we demonstrate that the proposed approach consistently outperforms other likelihood-based methods, thereby providing a mechanistic explanation for the effectiveness of our framework. The main contributions of our work are as follows:

- Under Gaussian perturbations, we derive entropy- and KL-divergence–based lower bounds on the difference between the expected log-likelihoods of ID and OOD samples (Theorems 3.1 and 4.1), which **explain how entropy manipulation can increase this gap** through controlled entropy increases.

- We propose a training-free algorithm, **Semantic Proportional Entropy Manipulation (SPEM)**, which scales Gaussian perturbations based on semantic similarity using a pretrained encoder and an ID memory bank, and performs OOD detection with the original flow likelihood, making SPEM a hybrid framework in which semantic similarity serves as an entropy manipulation signal rather than a direct OOD score. We further study a noise-only variant, SPEM-noise, and conduct experiments and analyses to explain the differences between SPEM and SPEM-noise.

- We conduct **extensive evaluations** across ten ID/OOD dataset pairs, including classical failure cases of flow-based methods, and observe almost consistent gains over likelihood-based baselines and recent multi-statistic methods, together with ablations on encoder choice, ReAct, and perturbation scaling.

## 2  Related Works

### 2.1  Normalizing Flow

Normalizing flows are a class of generative models, where input data $\mathbf{x} \in \mathbb{R}^d$ following distribution $P$ is transformed into $\mathbf{z} \in \mathbb{R}^d$, which typically follows a standard Gaussian distribution $\mathcal{N}(0, I_d)$ (Dinh et al., 2017), by using an invertible mapping $f : \mathbb{R}^d \to \mathbb{R}^d$ in order to estimate $P$. This generative model has the advantage that, through the change-of-variable formula, it can provide a tractable estimate of the likelihood without explicit knowledge of the true underlying distribution, which can be formally expressed as:

$$\log p_{\mathbf{x}}(\mathbf{x}) = \log p_{\mathcal{N}(0, I_d)}(\mathbf{z}) + \log \left| \det \frac{\partial \mathbf{z}}{\partial \mathbf{x}} \right|. \tag{2}$$

In a normalizing flow model, data generation proceeds by first sampling a latent vector $\mathbf{z}$ from the base distribution $\mathcal{N}(0, I_d)$. Then, the sampled latent vector is mapped into the data space through the inverse of the learned flow transformation $f^{-1}$. When designing a flow architecture, it is essential that the transformation function $f$ is invertible, and that the Jacobian determinant be efficiently computable. To satisfy these requirements, one approach is to construct transformations that are both easily invertible and have tractable Jacobian determinants (Rezende & Mohamed, 2015). Another widely used strategy is to employ coupling layers, in which the Jacobian takes a triangular form, thereby enabling efficient determinant computation (Dinh et al., 2015; 2017; Kingma & Dhariwal, 2018). In addition, Behrmann et al. (2019) proposed i-ResNet, constructing flows by enforcing Lipschitz constraints on residual networks to ensure invertibility. Building on this, Chen et al. (2019) addressed the issue of biased log-density estimation inherent in i-ResNet's approach while improving memory efficiency. In addition to these methodologies, a broad range of research efforts has investigated alternative designs to enhance the flexibility and expressiveness of normalizing flow. For instance, spline-based transformations have been proposed to provide more powerful and adaptive mapping functions within normalizing flows, thereby improving density estimation performance (Durkan et al., 2019).

Additionally, normalizing flows with stochastic sampling blocks relax topological constraints and achieve higher expressivity than deterministic flows (Wu et al., 2020).

## 2.2 Likelihood Paradox of Density Estimation Models

Nalisnick et al. (2019a) demonstrated that likelihood-based density estimation models such as normalizing flows can assign likelihoods in ways that contradict human intuition. For instance, when a flow is trained on CIFAR-10 as the ID and evaluated on SVHN as OOD, the resulting OOD detection performance is extremely poor, with AUROC values dropping below 10%. Building on this observation, Kirichenko et al. (2020) analyzed the underlying reasons in coupling-layer flows such as RealNVP and proposed methodological adjustments such as modifying masking strategies to alleviate the issue. Schirrmeister et al. (2020); Ren et al. (2019) improved OOD detection by training additional flow or background information and using the resulting likelihood ratio with the flows as the anomaly score. Serrà et al. (2020); Kamkari et al. (2024) proposed incorporating measures of input image complexity such as local intrinsic dimension or compression length using general-purpose compressors like PNG into the scoring function for improved OOD detection. Ahmadian et al. (2021); Morningstar et al. (2021); Osada et al. (2024) argued that relying solely on input data likelihood often leads to failures in OOD detection, and therefore proposed methods that incorporate additional statistics—such as latent likelihood, input complexity estimated via general-purpose compressors, and Jacobian determinants—within auxiliary classifiers trained to improve detection performance. Zhang et al. (2021); Le Lan & Dinh (2021) demonstrated that paradoxical behavior in likelihood assignment can arise even when a density estimation model perfectly estimates the ID. Choi et al. (2018) proposed an anomaly scoring method by ensembling generative models and employing the Watanabe-Akaike Information Criterion (WAIC) (Watanabe & Opper, 2010), and Nalisnick et al. (2019b) introduced an OOD detection approach based on the typicality set of ID data, which can outperform a likelihood-based detection approach. Caterini & Loaiza-Ganem (2022), which forms the foundation of Sections 3 and 4 of our paper, analyzed likelihood inversion and the effectiveness of likelihood ratio–based methods from an entropic perspective. Building on this entropic perspective, we analyze entropy manipulation and its effect on the expected log-likelihood gap between ID and OOD samples.

## 3 Can Entropy Manipulation Increase Performance?

In this section, we examine whether manipulating entropy can improve the performance of likelihood-based OOD detection. More specifically, in an oracle setting, we increase the entropy of OOD and examine how this affects the separation between ID and OOD log-likelihoods.

### 3.1 Problem Statement

We estimate the probability density function of the ID $P$ and aim to learn a model $P_\theta$ (e.g., a normalizing flow) that can estimate its likelihoods using only samples $\mathbf{x} \sim P$, assuming no access to OOD data during training. After training, $P_\theta$ is used to detect OOD samples $\mathbf{y} \sim Q \neq P$ based on their likelihoods. If the likelihood assigned by $P_\theta$ to a test vector $\mathbf{x}_{\text{test}}$ is below a threshold $\tau$, determined on ID, then $\mathbf{x}_{\text{test}}$ is classified as OOD. However, it has been reported that the majority of OOD data receive higher likelihood under $P_\theta$ than ID data, leading to failures of likelihood-based detection. In such cases, the following inequality generally holds:

$$\mathbb{E}_{\mathbf{x} \sim P}[\log P_\theta(\mathbf{x})] < \mathbb{E}_{\mathbf{x} \sim Q}[\log P_\theta(\mathbf{x})] \tag{3}$$

### 3.2 Entropy Manipulation

Previous studies have reported that the entropy (i.e., a proxy for complexity) of input data can affect the assignment of likelihood, and unintuitive behavior often arises when $\mathbb{H}(P) > \mathbb{H}(Q)$ (Caterini & Loaiza-Ganem, 2022; Serrà et al., 2020; Osada et al., 2024). Under the decomposition in Equation 1, the expected log-likelihood difference splits into two terms: $D_{KL}(Q \| P_\theta)$ and $\mathbb{H}(Q) - \mathbb{H}(P)$. Because $Q$ is unavailable during training, directly controlling $D_{KL}(Q \| P_\theta)$ is infeasible, whereas the entropy term can be increased

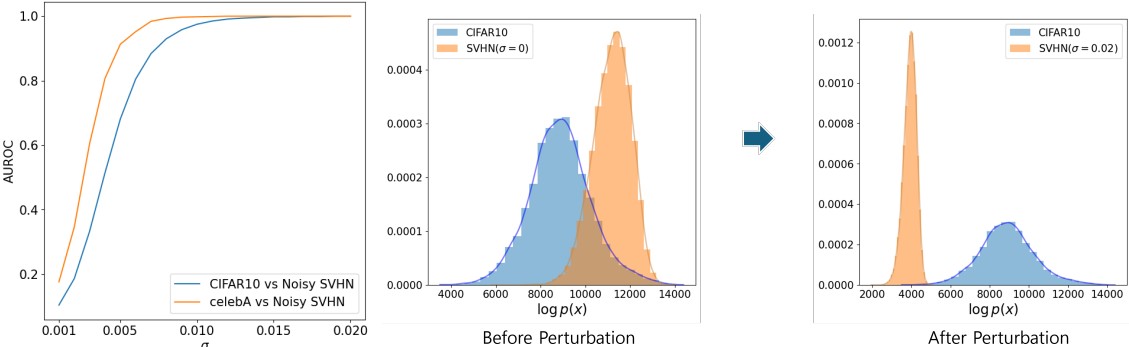

Figure 1: AUROC changes with entropy manipulation intensity and log-likelihood assignments with and without perturbation. We increase $\sigma$ from 0.001 to 0.02 and add $\mathbf{z} \sim N(0, \sigma^2 I_d)$ perturbation to SVHN to create a noisy SVHN distribution, and perform OOD detection through likelihood with Glow trained on CIFAR-10 and CelebA. The histogram shows the change in log-likelihood assignment before and after perturbation when Glow is trained with CIFAR-10.

by perturbing OOD samples to obtain a higher-entropy $Q'$. This insight raises the methodological question of whether manipulating entropy can systematically improve the performance of likelihood-based OOD detection:

**Question 1.** *If we increase OOD entropy via perturbation, wouldn't the expected log-likelihood difference align with intuition?*

Let $P$ be a continuous distribution on $\mathbb{R}^d$. For $X \sim P$, consider applying a Gaussian perturbation $Z \sim \mathcal{N}(0, \sigma^2 I_d)$. Then $X + Z$ follows $P' = P * Z$, and we expect $\mathbb{H}(P') > \mathbb{H}(P)$ because the perturbation increases uncertainty (see Appendix A for a formal proof of Theorem 3.1). Hence, we may be able to mitigate the likelihood inversion by increasing the entropy term of OOD that contributes to the expected likelihood difference.

To verify this hypothesis, we examine how detection performance changes as OOD entropy increases. We adopt settings known to exhibit likelihood inversion: CIFAR-10 or CelebA (Liu et al., 2015) image datasets as the ID and SVHN as the OOD dataset, training Glow (Kingma & Dhariwal, 2018) on each ID dataset. The ID/OOD pairs used in our experiments are those for which likelihood inversion has been reported in density estimation models, making them well-suited for testing our hypothesis (Kirichenko et al., 2020; Kamkari et al., 2024). We then added perturbations $Z \sim \mathcal{N}(0, \sigma^2 I_d)$ to SVHN samples and examined how OOD detection performance changes with different scales of $\sigma^2$. The corresponding experiment is shown in Figure 1, and the following observations were obtained.

**Observation 1.** *When detection fails in cases where $\mathbb{H}(OOD) < \mathbb{H}(ID)$, AUROC increases monotonically with the scale of Gaussian perturbation applied to the OOD.*

Figure 1 demonstrates that likelihood-based OOD detection yields low performance for very small $\sigma$, but the AUROC rapidly converges to 1 when $\sigma$ is greater than about 0.01. The histograms further indicate that perturbed log-likelihoods move in the intuitive direction at the sample level. These observations provide empirical evidence that increasing OOD entropy via perturbation can enhance likelihood-based detection. To explain why this improvement can occur, we derive Theorem 3.1 by extending Equation 1, and a formal proof is provided in Appendix A.

**Theorem 3.1.** *Let $P$, $P_\theta$, $Q$ be d-dimensional continuous probability distributions on $\mathbb{R}^d$. Let $X \sim Q$, $Z \sim \mathcal{N}(0, \sigma^2 I_d)$, and define $Q'$ as the distribution of $X + Z$. Then a lower bound on the expected log-likelihood difference estimated by $P_\theta$ between $P$ and $Q'$ is*

$$\frac{d}{2} \log\left(e^{\frac{2}{d}\mathbb{H}(X)} + 2\pi e\sigma^2\right) - \mathbb{H}(P) - D_{KL}(P||P_\theta).$$

By Theorem 3.1, the lower bound increases with the perturbation scale $\sigma$. This suggests that increasing the entropy gap, $\mathbb{H}(X + Z) - \mathbb{H}(P)$, helps the expected log-likelihood difference to align with our intuitive ordering. Note that interpreting performance variations purely as a function of the intensity of semantic information degradation is misleading; prior work shows that applying entropy-reducing transformations to the original distribution $Q$, such as collapsing inputs to a constant image, can worsen likelihood inversion (Osada et al., 2024).

## 4    Semantic Proportional Entropy Manipulation

In the previous section, we demonstrated that manipulating entropy through perturbations of the OOD data can improve OOD detection performance. However, at inference time we do not know whether a test sample comes from the ID or from OOD, so selectively perturbing only OOD samples is infeasible. Therefore, we consider a similarity-aware strategy that manipulates the perturbation strength using an ID memory bank. This raises a practical question:

**Question 2.** *Can we improve detection by increasing entropy more for low-similarity inputs than for high-similarity inputs, where similarity is measured by an ID memory bank?*

To analyze the effect of perturbations, we focus on the term $\mathbb{H}(Q) - \mathbb{H}(P) - D_{KL}(P||P_\theta)$, which forms part of the likelihood expectation difference under the setting $X \sim P$, $Y \sim Q$, and $P_\theta$ trained to estimate $P$. Assume that the KL-divergences among $P$, $Q$, and $P_\theta$ exist. Suppose we apply a weak perturbation $Z$ to $P$ and a stronger perturbation $Z'$ to $Q$, yielding perturbed distributions $X + Z \sim P'$ and $Y + Z' \sim Q'$. Under this setting, the lower bound for the gain in the likelihood expectation gap before and after applying perturbations to the ID and OOD can be derived as follows:

$$
\begin{aligned}
&\mathbb{E}_{\mathbf{x} \sim P'}[\log P_\theta(\mathbf{x})] - \mathbb{E}_{\mathbf{x} \sim Q'}[\log P_\theta(\mathbf{x})] \\
&- (\mathbb{E}_{\mathbf{x} \sim P}[\log P_\theta(\mathbf{x})] - \mathbb{E}_{\mathbf{x} \sim Q}[\log P_\theta(\mathbf{x})]) \\
&\geq (\mathbb{H}(Q') - \mathbb{H}(Q) - D_{KL}(Q||P_\theta)) \\
&- (\mathbb{H}(P') - \mathbb{H}(P) + D_{KL}(P'||P_\theta) - D_{KL}(P||P_\theta)).
\end{aligned}
\tag{4}
$$

According to Equation 4, if the entropy increase for the ID, together with the corresponding KL-divergence increment, $\mathbb{H}(P') - \mathbb{H}(P) + D_{KL}(P'||P_\theta) - D_{KL}(P||P_\theta)$, is smaller than the entropy increase for the OOD, $\mathbb{H}(Q') - \mathbb{H}(Q)$, then the lower bound of the gain in the likelihood expectation difference increases. The remaining issue is how to assign different perturbation strengths to ID and OOD samples. To this end, we propose **Semantic Proportional Entropy Manipulation (SPEM)**, a hybrid methodology that selectively adjusts perturbation intensity per sample using semantic similarity so that inputs with lower semantic similarity receive stronger perturbations than high-similarity inputs, where similarity serves as an entropy manipulation signal to bring likelihood ordering into alignment with human intuition, rather than a direct OOD score. This design manipulates OOD entropy efficiently without additional training, providing a lightweight mechanism to enhance likelihood-based OOD detection.

The proposed algorithm proceeds in three steps: first, given the ID dataset

$$
\mathcal{X}_{\text{in}} = \{\mathbf{x}_i\}_{i=1}^n,
$$

we train a density estimation model $f$, such as a normalizing flow, that provides tractable likelihoods for input data. Next, we construct a memory bank $\mathcal{M}$

$$
\mathcal{M} = \{h(\mathbf{x}_i)\}_{i=1}^n, \ \ s.t. \ h(\mathbf{x}_i) = \{\min(g(\mathbf{x}_i)_j, \beta)\}_{j=1}^{d'}
$$

where $g : \mathbb{R}^d \to \mathbb{R}^{d'}$ is a fixed feature extractor with publicly available pretrained weights on large-scale image data (e.g., ImageNet (Deng et al., 2009)), and $g(\mathbf{x}_i)_j$ denotes the $j$-th element of $g(\mathbf{x}_i)$. Each element of $\mathcal{M}$ stores the embedding vector of an ID sample. Additionally, we apply ReAct (Sun et al., 2021), which

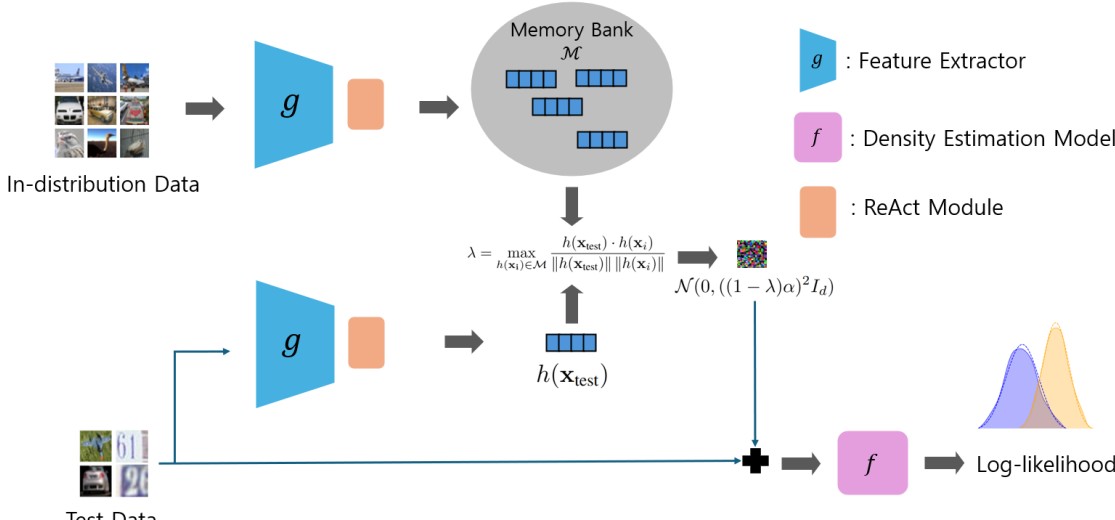

Figure 2: The overall framework of SPEM. $f$ is a density estimation model that provides a tractable likelihood for the input vector and estimates the ID, and $g$ is a feature extractor pretrained with general image data, capable of sufficiently extracting features for each image.

mitigates OOD overconfidence in the feature extractor, by rectifying the activation, which can be expressed as $h(\mathbf{x}_i)$ and $\beta$ is the limit of maximum activation determined by $p$-th quantile of activations. Through this process, the memory bank $\mathcal{M}$ stores compressed $d'$-dimensional embeddings rather than raw images, enabling an efficient estimate of ID proximity in representation space. Finally, for a test vector $\mathbf{x}_{\text{test}}$, we manipulate its entropy by applying the following Gaussian perturbation:

$$
\begin{aligned}
\mathbf{x}'_{\text{test}} &= \mathbf{x}_{\text{test}} + \mathcal{N}(0, ((1-\lambda)\alpha)^2 I_d), \\
s.t.\ \lambda &= \max_{h(\mathbf{x_i})\in\mathcal{M}} \frac{h(\mathbf{x}_{\text{test}}) \cdot h(\mathbf{x}_i)}{\|h(\mathbf{x}_{\text{test}})\|\, \|h(\mathbf{x}_i)\|}
\end{aligned}
\tag{5}
$$

where the perturbation intensity $\lambda$ is adaptively scaled according to the maximum cosine similarity between the test vector and the stored ID embeddings and $\alpha$ is a hyperparameter that controls the overall strength of entropy manipulation. Since the ID is assumed to contain multiple heterogeneous classes, we employ the maximum cosine similarity rather than the mean. Finally, we define the anomaly score as the log-likelihood estimated by the density estimation model $f$, and perform detection by classifying samples with lower likelihood values as OOD.

SPEM enables likelihood-based OOD detection to remain effective under both possible entropy orderings between ID and OOD, whereas existing likelihood-based scores often degrade in at least one of the two regimes (Osada et al., 2024). It uses the similarity $\lambda$, as an observable proxy of ID proximity to manipulate the amount of entropy increase at test time. Specifically, inputs with larger $\lambda$ are perturbed conservatively, whereas inputs with smaller $\lambda$ receive stronger perturbations. We then evaluate the test sample after this manipulation, and the final anomaly score is always computed from the flow log-likelihood of the perturbed input. In the inversion regime where OOD entropy is lower than ID entropy, this monotone scaling can suppress spuriously high likelihoods by increasing entropy more aggressively for low-similarity inputs, which helps improve separability. In the opposite regime where OOD entropy is higher than ID entropy, the same mechanism avoids unnecessarily disrupting high-similarity inputs and can further widen the effective discrepancy between the two distributions, leading to robust detection across both cases.

Furthermore, we formally derive how entropy manipulation through SPEM modifies the expected likelihood difference between ID and OOD samples under a density estimation model, and establish a lower bound on this difference as presented in Theorem 4.1. The proof of this theorem is reported in Appendix A.

**Theorem 4.1.** *Let $P$, $P_\theta$, and $Q$ be $d$-dimensional continuous probability distributions on $\mathbb{R}^d$. Let $X \sim P, Y \sim Q, Z \sim \mathcal{N}(0, \sigma_P^2 I_d), Z' \sim \mathcal{N}(0, \sigma_Q^2 I_d)$. Let $X + Z \sim P'$, $Y + Z' \sim Q'$. Let $P$ have covariance matrix $\Sigma$. Further assume that $\sigma_P$ and $\sigma_Q$ are positive random variables, each drawn independently from a continuous probability distribution supported on $(0, \infty)$. Then, the lower bound on the expected log-likelihood difference estimated by $P_\theta$ between $P'$ and $Q'$ is*

$$\frac{d}{2}\left(\log\left(\frac{e^{\frac{2}{d}\mathbb{H}(Y)} + 2\pi e(e^{\mathbb{E}[\log \sigma_Q^2]})}{2\pi e(\Pi_{i=1}^d(\lambda_i + \mathbb{E}[\sigma_P^2]))^{\frac{1}{d}}}\right)\right) - D_{KL}(P'||P_\theta)$$

*where $\lambda_i$ denotes the $i$-th eigenvalue of $\Sigma$.*

Theorem 4.1 demonstrates that, if the perturbation applied to $Q$ is sufficiently stronger than that applied to $P$ (i.e., generally $\sigma_Q > \sigma_P$), then the lower bound of the expected log-likelihood difference can be increased. Additionally, if the feature extractor $g$ is well trained, the similarity between $g(\mathbf{x}_{\text{test}})$ and the ID embeddings stored in the memory bank naturally yields a smaller perturbation scale for high-similarity inputs than for low-similarity inputs. Consequently, Theorem 4.1 provides theoretical evidence that SPEM enlarges the expected log-likelihood difference between ID and OOD.

## 5 Experiment

In this section, we compare the OOD detection performance of SPEM on real world datasets against other likelihood-based approaches.

**Data Preprocessing** We utilized CIFAR-10, CIFAR-100, SVHN, CelebA, MNIST (Deng, 2012), and FashionMNIST (Xiao et al., 2017) datasets provided by torchvision (Marcel & Rodriguez, 2010), which are widely adopted benchmark datasets in likelihood-based OOD detection methods. From these, we constructed five dataset pairs, each consisting of one ID dataset and one OOD dataset, and we also evaluated the reversed pairing. In total, we evaluated OOD detection performance across ten ID/OOD dataset pairs. All images were resized to $32 \times 32$, and the train/test splits provided by torchvision were adopted for each dataset. We applied uniform dequantization by adding pixel-wise noise $u \sim U(0, 1/256)$ to the inputs, enabling them to be interpreted as continuous distributions. Since we employed feature extractors pretrained on ImageNet, which generally take three channels images as input, we converted MNIST and FashionMNIST to three-channel format by replicating the single channel across all three channels. In our experiments, the detection performance was measured using AUROC.

**Likelihood-based Models** All comparison models except the local intrinsic dimension-based method are based on ResFlow, a normalizing flow model commonly used in the image domain, with latent distribution $\mathcal{N}(0, I_d)$, and we compare our proposed method with the following seven approaches:

1. Likelihood only. For a test vector $\mathbf{x}$, we directly use negative log-likelihood $S(\mathbf{x}) = -\log p(\mathbf{x})$ as the anomaly score for OOD detection.

2. Complexity-based method (Serrà et al., 2020). We measure the bit length $L(\mathbf{x})$ by compressing the image using PNG and obtaining the resulting number of bits. The OOD score is then defined as $S(\mathbf{x}) = -\log p(\mathbf{x}) - L(\mathbf{x})$, which incorporates the complexity term into the likelihood.

3. Typicality-based method ($\sqrt{d}$) (Nalisnick et al., 2019b). Since the latent of the normalizing flow is assumed to follow $\mathcal{N}(0, I_d)$, when the ID is $P$, we use $S(\mathbf{z}') = |\sqrt{d} - ||\mathbf{z}'||^2|$ as the OOD score, leveraging high-dimensional Gaussian concentration around a thin shell; thus $S(\mathbf{z}')$ measures how far the latent $\mathbf{z}'$ deviates from this typical set.

4. Typicality-based method (Entropy) (Nalisnick et al., 2019b). Similar to typicality test using latent, we define $S(\mathbf{x}') = |\mathbb{E}_P[\log p(\mathbf{x})] - \log p(\mathbf{x}')|$ as OOD score, which measures how far the input data $\mathbf{x}'$ falls from the input data's typicality set.

Table 1: OOD detection performance on real image dataset using ResFlow. The five dataset pairs on the left are widely recognized as cases where likelihood-based OOD detection fails, and the remaining pairs do not exhibit problematic behavior in the estimated likelihood. For each dataset pair, the values that ranked within the top two across all compared models (excluding the $\lambda$) are highlighted in bold. Results for SPEM and SPEM-noise are reported as the mean AUROC over three independent random seeds.

| $P$ (In) | CIFAR-10 | CIFAR-100 | CelebA | CIFAR-10 | FashionMNIST | SVHN | SVHN | SVHN | CelebA | MNIST |
|---|---|---|---|---|---|---|---|---|---|---|
| $Q$ (Out) | SVHN | SVHN | SVHN | CelebA | MNIST | CIFAR-10 | CIFAR-100 | CelebA | CIFAR-10 | FashionMNIST |
| Likelihood | 0.0256 | 0.0277 | 0.0163 | 0.6346 | 0.8797 | 0.9967 | 0.9950 | 0.9988 | 0.5819 | 0.9786 |
| Complexity | 0.7943 | 0.7331 | 0.7998 | 0.6252 | 0.8811 | 0.9912 | 0.9890 | 0.9983 | 0.5740 | 0.3411 |
| Typicality ($\sqrt{d}$) | 0.0226 | 0.0273 | 0.0306 | 0.5429 | 0.7914 | 0.9969 | 0.9954 | 0.9987 | 0.6628 | 0.9954 |
| Typicality (Entropy) | 0.8866 | 0.8783 | 0.9489 | 0.3633 | 0.7703 | 0.9938 | 0.9893 | 0.9987 | 0.6082 | 0.9688 |
| Likelihood Ratio | 0.8512 | 0.7952 | 0.4458 | 0.7018 | **0.9718** | 0.9640 | 0.9418 | 0.9962 | 0.1826 | 0.0874 |
| GMM | 0.8916 | 0.8546 | 0.9328 | 0.4050 | 0.8486 | 0.9930 | 0.9899 | 0.9962 | 0.6845 | 0.9919 |
| LID | 0.9360 | 0.9330 | 0.9490 | 0.6550 | **0.9510** | 0.9870 | 0.9860 | 0.9960 | 0.9390 | **1.0000** |
| SPEM | **0.9915** | **0.9359** | 1.0000 | **0.9830** | 0.9206 | **0.9997** | **0.9993** | 1.0000 | **0.9999** | 1.0000 |
| SPEM-noise | **0.9942** | **0.9460** | 1.0000 | **0.9876** | 0.9440 | **0.9997** | 0.9989 | 1.0000 | **0.9999** | 1.0000 |
| $\lambda$ (Similarity-only) | 0.9954 | 0.9530 | 1.0000 | 0.9894 | 0.9445 | 0.9998 | 0.9990 | 1.0000 | 0.9999 | 1.0000 |

5. Likelihood ratio method (Ren et al., 2019). We define the OOD score as the likelihood ratio score $S(\mathbf{x}) = -(\log p_\theta(\mathbf{x}) - \log p_{\theta_b}(\mathbf{x}))$, where $p_\theta$ denotes the model trained on the ID and $p_{\theta_b}$ is the background-trained model constructed by introducing perturbations at the pixel level.

6. GMM method using various statistics (Osada et al., 2024). We train a Gaussian Mixture Model (GMM) using the latent log-likelihood $\log p(\mathbf{z})$ and the bit-length $L(\mathbf{x})$ of the input data $\mathbf{x}$ obtained from a general compressor such as PNG, and employ the negative log-likelihood estimated by the GMM as the OOD score.

7. Local intrinsic dimension-based method Kamkari et al. (2024). This method performs OOD detection using a dual threshold based on local intrinsic dimension (LID) and log-likelihood. While the original formulation requires computing the Jacobian of the flow, the computational cost makes a faithful re-implementation impractical. To ensure comparability, we therefore report the AUROC values provided in the original paper. Since the official implementation of LID relies on a validation set and a slightly different data-split protocol, we treat the reported AUROC as a reference upper bound rather than a strictly comparable baseline. Nevertheless, including LID in our tables enables a fairer contextualization against established state-of-the-art methods.

SPEM uses a pretrained ResNet-152 (He et al., 2016) feature extractor provided by torchvision (Marcel & Rodriguez, 2010) and the hyperparameter $\alpha$, which controls the manipulation intensity, was set to 0.4 in all experimental settings. The feature extractor is kept fixed (no training or fine-tuning on ID/OOD data) and is used only to compute embeddings for scaling the perturbation magnitude. Additionally, to examine the influence of the semantic information contained in the original images, we introduced **SPEM-noise** as a comparative model, which uses only the noise derived through similarity as input to the density model. In this setting, the hyperparameter $\alpha$ was fixed at 0.1. We also report the results of using the similarity $\lambda$ alone for OOD detection (i.e., without likelihood evaluation), to assess how much separability is provided by the similarity itself.

**Experiment Result** According to Table 1, SPEM improves substantially over raw likelihood on the widely recognized failure pairs, and remains competitive on the remaining pairs where likelihood-based scoring is already near-saturated. More broadly, SPEM is the only method (aside from LID, which is not directly comparable in our setting) that maintains consistently high AUROC across both inversion and non-inversion regimes. Interestingly, we find that SPEM-noise, which does not incorporate the original test data, outperforms SPEM in some hard cases. This result implies that strong detection can be achieved even when the original test content is removed, since SPEM-noise still induces a well-controlled entropy term through Gaussian inputs whose scale is adjusted by similarity. This suggests that the likelihood ordering is not primarily determined by fine-grained semantics of the original images; rather, the entropy of the original ID/OOD distributions can act as a confounder in likelihood-based ordering. To further substantiate this

Table 2: OOD detection performance (AUROC) comparison between SPEM and $k$-NN (Sun et al., 2022) across all ten ID/OOD dataset pairs. $k$-NN is evaluated in the same pretrained ResNet-152 feature space as SPEM, serving as the most natural label-free embedding-based baseline for a fair feature-space comparison.

| $P$ (In) | CIFAR-10 | CIFAR-100 | CelebA | CIFAR-10 | FashionMNIST | SVHN | SVHN | SVHN | CelebA | MNIST |
| $Q$ (Out) | SVHN | SVHN | SVHN | CelebA | MNIST | CIFAR-10 | CIFAR-100 | CelebA | CIFAR-10 | FashionMNIST |
|---|---|---|---|---|---|---|---|---|---|---|
| SPEM | **0.9913** | **0.9359** | **1.0000** | 0.9830 | 0.9207 | 0.9997 | **0.9992** | **1.0000** | **0.9999** | **1.0000** |
| $k$-NN | 0.9712 | 0.8668 | **1.0000** | **0.9934** | **0.9450** | **0.9999** | 0.9989 | **1.0000** | **0.9999** | 0.9999 |

observation, we include in Appendix C an ablation study that analyzes the impact of the feature extractor used in SPEM, the effectiveness of ReAct in refining the embedding vector, and additional experiments on alternative flow models to assess the generality of our findings beyond ResFlow.

We additionally compare SPEM against $k$-NN (Sun et al., 2022), a feature similarity-based OOD detection method using the same pretrained feature space. Methods such as Mahalanobis distance (Lee et al., 2018) require class label information, and logit-based approaches including MSP (Hendrycks & Gimpel, 2017), MaxLogit (Hendrycks et al., 2022), and Energy (Liu et al., 2020) rely on a supervised classifier's output layer. As SPEM operates in a fully unsupervised setting using only a training-free pretrained ImageNet encoder, these methods fall outside our comparison scope. We therefore include $k$-NN in the same pretrained feature space as the most natural label-free embedding-based baseline, and report results across all ten dataset pairs in Table 2. As shown in Table 2, SPEM matches or outperforms $k$-NN on 7 out of 10 dataset pairs, demonstrating that entropy manipulation through flow likelihood provides genuine contribution beyond feature-similarity alone. On the two inversion-regime pairs where $\mathbb{H}(P) > \mathbb{H}(Q)$ (CIFAR-10 vs. CelebA and FashionMNIST vs. MNIST), $k$-NN outperforms SPEM with a noticeable margin. This is theoretically consistent: in the inversion regime the flow likelihood faces a larger entropic challenge and while SPEM mitigates this through entropy manipulation the similarity-based signal is inherently unaffected by the entropy ordering between ID and OOD.

Table 3: OOD detection performance of SPEM and sampled $\lambda$. Results are reported as the mean AUROC over three repeated samplings. We use ResFlow as the flow model and set $\alpha = 0.1$.

| $P$ (In) | | SVHN | |
| $Q$ (Out) | CIFAR-10 | CIFAR-100 | CelebA |
|---|---|---|---|
| SPEM | **0.8159** | **0.8190** | **0.7846** |
| $\lambda$ | 0.7591 | 0.7598 | 0.7603 |

Although SPEM slightly underperforms the similarity-only baseline $\lambda$ on some hard pairs and is comparable to it on easy pairs, one might reasonably interpret $\lambda$-only detection as an empirical ceiling for SPEM in such regimes. This ceiling can appear when $\lambda$ already offers strong separability and the AUROC is near saturation, making further improvements difficult to observe. In Table 1, the similarity-only score $\lambda$ is near-saturated on several pairs, making it hard to assess whether SPEM improves beyond similarity itself. To avoid this saturation, we run a controlled similarity test where $\lambda$ has only moderate discriminative power: we draw $\lambda_{\text{ID}} \sim \mathcal{N}(0.65, 0.05^2)$ and $\lambda_{\text{OOD}} \sim \mathcal{N}(0.60, 0.05^2)$, clipped to $[0, 1]$, which yields AUROC $\approx 0.75$ for $\lambda$-only. We then use these $\lambda$ values only to set SPEM's perturbation scale, while scoring with the flow log-likelihood on perturbed inputs. As shown in Table 3, SPEM consistently outperforms the $\lambda$-only baseline across all three settings, indicating that $\lambda$-only is not a strict ceiling for SPEM when similarity is imperfect. These results correspond to a regime that is complementary to SPEM-noise.

Furthermore, we conduct an experiment on a realistic near-OOD setting, where MNIST is used as the ID and MNIST-C (Mu & Gilmer, 2019), a benchmark consisting of MNIST images with various corruptions applied, is used as the OOD. We compare the performance of SPEM and $\lambda$-only across all corruption types, using ResNet-152 as the feature extractor and $\alpha = 0.4$. The results are reported in Table 4. SPEM consistently matches or outperforms $\lambda$-only on corruption types that induce meaningful entropy changes

Table 4: OOD detection performance on MNIST-C with MNIST as the ID dataset. We evaluate SPEM and $\lambda$-only across 15 corruption types and report AUROC for each case.

| | brightness | canny | dotted | fog | glass | impulse | motion | rotate | scale | shear | shot | spatter | stripe | translate | zigzag |
|---|---|---|---|---|---|---|---|---|---|---|---|---|---|---|---|
| SPEM ($\alpha = 0.4$) | **0.9917** | **0.9903** | **0.9615** | **0.9999** | **0.9997** | **0.9999** | **0.9972** | 0.6179 | 0.7679 | **0.6982** | 0.9929 | **0.9909** | **0.9993** | 0.7031 | **0.9690** |
| $\lambda$ | 0.9826 | 0.9896 | 0.9589 | 0.9997 | **0.9997** | **0.9999** | 0.9968 | **0.6201** | **0.7882** | 0.6977 | **0.9930** | 0.9904 | 0.9987 | **0.7067** | 0.9662 |

(e.g., fog, blur, noise-based corruptions), while $\lambda$-only outperforms SPEM on geometric transformations (e.g., rotate, scale, translate) that largely preserve the entropy of the original distribution. This pattern is precisely what our theory predicts: entropy manipulation through the flow likelihood provides a genuine independent contribution beyond the similarity signal when the entropy gap between ID and OOD is meaningful, and its contribution diminishes otherwise.

In those case, retaining the original input distribution and its inherent entropy appears to contribute to likelihood separation. From a practitioner's perspective, if one can reasonably hypothesize whether the OOD entropy is higher or lower than the ID entropy based on domain knowledge or prior deployments, then choosing an appropriate perturbation scale $\alpha$ provides a simple knob to further improve detection in the corresponding regime. However, when such prior information is unavailable at test time, our similarity-conditioned approach remains effective regardless of the entropy ordering between ID and OOD.

## 6    Analysis of SPEM-noise

In Tables 1 and 9, SPEM-noise matches or even surpasses SPEM in several settings. Unlike SPEM, SPEM-noise ignores the test input and evaluates the density model on Gaussian noise whose scale is calibrated by the maximum similarity between the test embedding and the ID memory bank. Despite containing no semantic content, this procedure can yield comparable or superior OOD detection performance. To explain this behavior, we analyze how SPEM and SPEM-noise change the expected log-likelihood gap between in- and out-of-distribution samples.

Let $X \sim P$ (ID) and $Y \sim Q$ (OOD) with covariance $\Sigma_Q$. Let $Z \sim \mathcal{N}(0, \sigma_P^2 I_d)$ and $Z' \sim \mathcal{N}(0, \sigma_Q^2 I_d)$ denote the noises applied to $P$ and $Q$, and write $X + Z \sim P'$ and $Y + Z' \sim Q'$ for the perturbed distributions under the density model $P_\theta$. While $\sigma_P^2$ and $\sigma_Q^2$ depend on similarity in practice, we treat them as constants here for analytical tractability. We then define $\Delta$ as the increment in the log-likelihood difference under SPEM-noise relative to SPEM, and decompose it into an entropy increment $\Delta_E$ and a KL-divergence increment $\Delta_{KL}$. We first establish conditions for $\Delta_E > 0$ and characterize its dependence on the noise strength in Theorem 6.1 and Theorem 6.2.

**Theorem 6.1.** *Let $P$, $P_\theta$, and $Q$ be $d$-dimensional continuous probability distributions on $\mathbb{R}^d$. Let*

$$X \sim P, \quad Y \sim Q, \quad Z \sim \mathcal{N}(0, \sigma_P^2 I_d), \quad Z' \sim \mathcal{N}(0, \sigma_Q^2 I_d),$$

*and define*

$$X + Z \sim P', \quad Y + Z' \sim Q'.$$

*Let $Q$ have covariance matrix $\Sigma_Q$. Then, the sufficient condition of $\Delta_E > 0$ is*

$$\frac{\sigma_Q^2}{\sigma_P^2} > \frac{2\pi e \left( tr(\Sigma_Q) \right)}{d e^{\frac{2}{d} \mathbb{H}(X)}}.$$

**Theorem 6.2.** *Let $P$, $P_\theta$, and $Q$ be $d$-dimensional continuous probability distributions on $\mathbb{R}^d$. Let*

$$X \sim P, \quad Y \sim Q, \quad Z \sim \mathcal{N}(0, \sigma_P^2 I_d), \quad Z' \sim \mathcal{N}(0, \sigma_Q^2 I_d),$$

*and define*

$$X + Z \sim P', \quad Y + Z' \sim Q'.$$

*Assume the Fisher information of $X$ and $Y$ are non-zero finite. Then, $\frac{\partial \Delta_E}{\partial \sigma_P^2} < 0$ and $\frac{\partial \Delta_E}{\partial \sigma_Q^2} > 0$.*

Therefore, by Theorems 6.1 and 6.2, we establish that $\Delta_E$ not only becomes strictly positive once the noise variance ratio $\sigma_Q^2/\sigma_P^2$ exceeds a certain threshold, but also exhibits opposite monotonic behaviors with respect to the ID/OOD noise variances. Specifically, $\Delta_E$ exhibits a negative gradient with respect to $\sigma_P^2$ and a positive gradient with respect to $\sigma_Q^2$. Consequently, by decreasing $\sigma_P^2$ while simultaneously increasing $\sigma_Q^2$ such that their ratio surpasses the identified threshold, one can guarantee that $\Delta_E$, as a constituent component of $\Delta$, remains positive and strictly increases.

Since the overall increment $\Delta$ is influenced by $\Delta_{\mathrm{KL}}$, an analysis of $\Delta_{\mathrm{KL}}$ is required in order to explain the expected log-likelihood under both SPEM-noise and SPEM. However, unlike entropy, $\Delta_{\mathrm{KL}}$ is difficult to bound due to the arbitrariness of the OOD setting. Therefore, we relax this condition and, in Theorem 6.3, derive the bound under the realistic assumptions that the negative log-likelihood is $L$-smooth and $\lambda$-semiconvex, in order to analyze the bound of the overall difference $\Delta$. These conditions accommodate weakly multi-modal distributions and are more practically plausible than assuming an $L$-Lipschitz log-likelihood, as required in Theorem C.11.

**Theorem 6.3.** *Let $P$, $P_\theta$, and $Q$ be $d$-dimensional continuous probability distributions on $\mathbb{R}^d$ and have finite first and second moments. Let*

$$X \sim P, \quad Y \sim Q, \quad Z \sim \mathcal{N}(0, \sigma_P^2 I_d), \quad Z' \sim \mathcal{N}(0, \sigma_Q^2 I_d),$$

*and define*

$$X + Z \sim P', \quad Y + Z' \sim Q'.$$

*Let $f(\mathbf{x}) = -\log P_\theta(\mathbf{x})$ be $\lambda$-semiconvex and $L$-smooth. Then*

$$\Delta \in [-C - \frac{d(\lambda + L)}{2}(\sigma_P^2 + \sigma_Q^2), -C + \frac{d(\lambda + L)}{2}(\sigma_P^2 + \sigma_Q^2)]$$

*where $C = \mathbb{E}_{\mathbf{x} \sim P}[\log P_\theta(\mathbf{x})] - \mathbb{E}_{\mathbf{x} \sim Q}[\log P_\theta(\mathbf{x})]$. Also, a sufficient condition for $\Delta > 0$ is $-\frac{2C}{d(\lambda + L)} > \sigma_P^2 + \sigma_Q^2$.*

By Theorem 6.3, we establish the existence of a guaranteed lower bound when the negative log-likelihood satisfies the $L$-smoothness and $\lambda$-semiconvexity conditions. This existence result holds when $C$ is negative, that is, when the log-likelihood expectation of the OOD exceeds that of the ID. Consequently, this suggests that SPEM-noise may achieve superior OOD detection performance compared to SPEM in certain cases. For future work, it would be of interest to relax the assumption and investigate the effectiveness of SPEM-noise under weaker conditions, and to analyze the conditions under which $\Delta_{KL}$ becomes positive or takes values smaller than $\Delta_E$. Such analyses would further explain when and why SPEM-noise can yield improved OOD detection performance. Detailed analyses of SPEM-noise and proofs of the theorems are provided in Appendix C.7.

## 7 Conclusion

We proposed SPEM to mitigate the likelihood paradox arising from likelihood assignment in generative models. From an entropic perspective, we explained why SPEM succeeds not only in cases where conventional likelihood-based detection methods fail in OOD detection but also in scenarios where they already perform well. Furthermore, we empirically demonstrated on real-world datasets that the complementary effect of entropy manipulation and semantic similarity in SPEM outperforms existing baselines that rely solely on generative model likelihoods, and achieves performance comparable to feature-space baselines such as $k$-NN. Since both our method and other approaches in this research field still require auxiliary processes, we anticipate future work to develop training methodologies that align the likelihood assignment of generative models with human intuition, enabling likelihood-based OOD detection without auxiliary procedures, as well as methods that further close the gap with embedding-based approaches by more effectively leveraging the flow likelihood. In SPEM specifically, the degree of entropy manipulation is governed by the pretrained similarity signal $\lambda$, and we identify developing entropy manipulation strategies that are independent of such similarity-based signals as a promising direction for future work. Through our work, we hope this study will stimulate further investigation into how to interpret and utilize likelihood estimates produced by generative models.

**Broader Impact Statement**

This work advances the methodological understanding of likelihood-based OOD detection in generative models. By clarifying the role of entropy in likelihood inversion and introducing a training-free test-time procedure, the proposed approach can improve the reliability of likelihood-based OOD detection under distributional shift, with potential relevance to anomaly detection and monitoring settings. We do not anticipate direct negative societal impacts from this work; however, as with other machine learning methods, likelihood-based models should be applied with caution in high-stakes scenarios without sufficient validation.

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

## A  Proof of Theorems

**Theorem A.1.** *Let $P$, $P_\theta$, $Q$ be d-dimensional continuous probability distributions on $\mathbb{R}^d$. Let $X \sim Q$, $Z \sim \mathcal{N}(0, \sigma^2 I_d)$, and define $Q'$ as the distribution of $X + Z$. Then a lower bound on the expected log-likelihood difference estimated by $P_\theta$ between $P$ and $Q'$ is*

$$\frac{d}{2}\log\left(e^{\frac{2}{d}\mathbb{H}(X)} + 2\pi e \sigma^2\right) - \mathbb{H}(P) - D_{KL}(P||P_\theta).$$

*Proof.* The difference between the log-likelihood expectations estimated by $P_\theta$ for $P$ and $Q$ can be derived as follows:

$$
\begin{aligned}
&\mathbb{E}_{\mathbf{x} \sim P}[\log P_\theta(\mathbf{x})] - \mathbb{E}_{\mathbf{x} \sim Q}[\log P_\theta(\mathbf{x})] \\
&= -D_{KL}(P||P_\theta) - \mathbb{H}(P) + D_{KL}(Q||P_\theta) + \mathbb{H}(Q) \\
&= D_{KL}(Q||P_\theta) - D_{KL}(P||P_\theta) + \mathbb{H}(Q) - \mathbb{H}(P)
\end{aligned}
\tag{6}
$$

Therefore, if we rewrite this equation by modifying $Q$ to $Q'$,

$$
\begin{aligned}
&\mathbb{E}_{\mathbf{x} \sim P}[\log P_\theta(\mathbf{x})] - \mathbb{E}_{\mathbf{x} \sim Q'}[\log P_\theta(\mathbf{x})] \\
&= D_{KL}(Q'||P_\theta) - D_{KL}(P||P_\theta) + \mathbb{H}(Q') - \mathbb{H}(P) \\
&\geq -D_{KL}(P||P_\theta) + \mathbb{H}(Q') - \mathbb{H}(P)
\end{aligned}
\tag{7}
$$

If we express $\mathbb{H}(Q')$ in terms of a random variable rather than a distribution, we get the following:

$$\mathbb{H}(Q') = \mathbb{H}(X + Z) \tag{8}$$

The entropy power of $X$ is defined as (Cover, 1999):

$$N(X) = \frac{1}{2\pi e} e^{\frac{2}{d}\mathbb{H}(X)} \tag{9}$$

Since $X$ and $Z$ are independent, the following formula holds due to entropy power inequality:

$$\begin{aligned}
&\frac{1}{2\pi e} e^{\frac{2}{d}\mathbb{H}(X+Z)} \geq \frac{1}{2\pi e}(e^{\frac{2}{d}\mathbb{H}(X)} + e^{\frac{2}{d}\mathbb{H}(Z)}) \\
&\Rightarrow e^{\frac{2}{d}\mathbb{H}(X+Z)} \geq e^{\frac{2}{d}\mathbb{H}(X)} + e^{\frac{2}{d}\mathbb{H}(Z)} \\
&\Rightarrow \mathbb{H}(X+Z) \geq \frac{d}{2}\log(e^{\frac{2}{d}\mathbb{H}(X)} + 2\pi e \sigma^2)
\end{aligned} \tag{10}$$

Therefore, we can derive as follows:

$$\begin{aligned}
&\mathbb{E}_{\mathbf{x} \sim P}[\log P_\theta(\mathbf{x})] - \mathbb{E}_{\mathbf{x} \sim Q'}[\log P_\theta(\mathbf{x})] \\
&\geq -D_{KL}(P||P_\theta) + \mathbb{H}(Q') - \mathbb{H}(P) \\
&\geq \frac{d}{2}\log(e^{\frac{2}{d}\mathbb{H}(X)} + 2\pi e \sigma^2) - \mathbb{H}(P) - D_{KL}(P||P_\theta)
\end{aligned} \tag{11}$$

$\square$

**Theorem A.2.** *Let $P$, $P_\theta$, and $Q$ be $d$-dimensional continuous probability distributions on $\mathbb{R}^d$. Let $X \sim P, Y \sim Q, Z \sim \mathcal{N}(0, \sigma_P^2 I_d), Z' \sim \mathcal{N}(0, \sigma_Q^2 I_d)$. Let $X + Z \sim P'$, $Y + Z' \sim Q'$. Let $P$ have covariance matrix $\Sigma$. Further assume that $\sigma_P$ and $\sigma_Q$ are positive random variables, each drawn independently from a continuous probability distribution supported on $(0, \infty)$. Then, the lower bound on the expected log-likelihood difference estimated by $P_\theta$ between $P'$ and $Q'$ is*

$$\frac{d}{2}\left(\log\left(\frac{e^{\frac{2}{d}\mathbb{H}(Y)} + 2\pi e(e^{\mathbb{E}[\log \sigma_Q^2]})}{2\pi e(\Pi_{i=1}^d(\lambda_i + \mathbb{E}[\sigma_P^2]))^{\frac{1}{d}}}\right)\right) - D_{KL}(P'||P_\theta)$$

*where $\lambda_i$ denotes the $i$-th eigenvalue of $\Sigma$.*

*Proof.* Since $Y$ and $Z'$ are independent, the following formula holds due to entropy power inequality:

$$\mathbb{H}(Q') = \mathbb{H}(Y + Z') \geq \frac{d}{2}\log(e^{\frac{2}{d}\mathbb{H}(Y)} + e^{\frac{2}{d}\mathbb{H}(Z')}) \tag{12}$$

We can expand $\mathbb{H}(Z')$ as follows using the property of conditional entropy:

$$\begin{aligned}
\mathbb{H}(Z') &= \mathbb{E}\left[\mathbb{H}(Z'|\sigma_Q)\right] + \mathcal{I}(Z'; \sigma_Q) \\
&= \frac{d}{2}(\log(2\pi e) + \mathbb{E}[\log \sigma_Q^2]) + \mathcal{I}(Z'; \sigma_Q) \; (\because \mathbb{H}(Z'|\sigma_Q) = \frac{d}{2}\log(2\pi e \sigma_Q^2))
\end{aligned} \tag{13}$$

Therefore, we can rewrite lower bound of $\mathbb{H}(Y + Z')$ as follows:

$$\mathbb{H}(Y + Z') \geq \frac{d}{2} \log(e^{\frac{2}{d}\mathbb{H}(Y)} + e^{\frac{2}{d}\mathbb{H}(Z')})$$

$$= \frac{d}{2} \log(e^{\frac{2}{d}\mathbb{H}(Y)} + e^{\log(2\pi e) + \mathbb{E}[\log \sigma_Q^2] + \frac{2}{d}\mathcal{I}(Z';\sigma_Q)})$$

$$\geq \frac{d}{2} \log(e^{\frac{2}{d}\mathbb{H}(Y)} + e^{\log(2\pi e) + \mathbb{E}[\log \sigma_Q^2]}) \quad (\because \ \mathcal{I}(Z';\sigma_Q) \geq 0)$$

$$= \frac{d}{2} \log(e^{\frac{2}{d}\mathbb{H}(Y)} + 2\pi e(e^{\mathbb{E}[\log \sigma_Q^2]}))$$

(14)

Hence, we can derive:

$$\mathbb{E}_{\mathbf{x} \sim P'}[\log P_\theta(\mathbf{x})] - \mathbb{E}_{\mathbf{x} \sim Q'}[\log P_\theta(\mathbf{x})]$$

$$\geq -D_{KL}(P'||P_\theta) + \mathbb{H}(Q') - \mathbb{H}(P')$$

$$= \mathbb{H}(Y + Z') - \mathbb{H}(P') - D_{KL}(P'||P_\theta)$$

$$= \frac{d}{2} \log(e^{\frac{2}{d}\mathbb{H}(Y)} + 2\pi e(e^{\mathbb{E}[\log \sigma_Q^2]})) - \mathbb{H}(P') - D_{KL}(P'||P_\theta)$$

(15)

Since $X$ and $Z$ are independent, the following holds due to the maximum entropy property of Gaussian distribution:

$$\mathbb{H}(P') \leq \mathbb{H}(\mathcal{N}(0, \text{Cov}(X + Z))$$

$$= \mathbb{H}(\mathcal{N}(0, \text{Cov}(X) + \text{Cov}(Z))$$

(16)

Because of law of total covariance, we can derive:

$$\mathbb{H}(P') \leq \mathbb{H}(\mathcal{N}(0, \text{Cov}(X + Z)))$$

$$= \mathbb{H}(\mathcal{N}(0, \text{Cov}(X) + \text{Cov}(Z))$$

$$= \mathbb{H}(\mathcal{N}(0, \Sigma + \mathbb{E}[\text{Cov}(Z|\sigma_P)] + \text{Cov}(\mathbb{E}[Z|\sigma_P])) \quad (\because \text{Law of Total Covariance})$$

$$= \mathbb{H}(\mathcal{N}(0, \Sigma + \mathbb{E}[\sigma_P^2]I_d)) \quad (\because \mathbb{E}[Z|\sigma_P] = 0, \text{Cov}(Z|\sigma_P) = \sigma_P^2 I_d)$$

$$= \frac{1}{2} \log((2\pi e)^d \det(\Sigma + \mathbb{E}[\sigma_P^2]I_d))$$

(17)

Therefore, we can derive as follows:

$$\mathbb{E}_{\mathbf{x} \sim P'}[\log P_\theta(\mathbf{x})] - \mathbb{E}_{\mathbf{x} \sim Q'}[\log P_\theta(\mathbf{x})]$$

$$\geq \frac{d}{2} \left( \log(e^{\frac{2}{d}\mathbb{H}(Y)} + 2\pi e(e^{\mathbb{E}[\log \sigma_Q^2]})) \right) - \mathbb{H}(P') - D_{KL}(P'||P_\theta)$$

$$\geq \frac{d}{2} \left( \log(e^{\frac{2}{d}\mathbb{H}(Y)} + 2\pi e(e^{\mathbb{E}[\log \sigma_Q^2]})) \right) - \frac{1}{2} \log((2\pi e)^d \det(\Sigma + \mathbb{E}[\sigma_P^2]I_d)) - D_{KL}(P'||P_\theta)$$

$$= \frac{d}{2} \left( \log(e^{\frac{2}{d}\mathbb{H}(Y)} + 2\pi e(e^{\mathbb{E}[\log \sigma_Q^2]})) \right) - \frac{d}{2} \log(2\pi e(\det(\Sigma + \mathbb{E}[\sigma_P^2]I_d))^{\frac{1}{d}}) - D_{KL}(P'||P_\theta)$$

$$= \frac{d}{2} \left( \log(e^{\frac{2}{d}\mathbb{H}(Y)} + 2\pi e(e^{\mathbb{E}[\log \sigma_Q^2]})) - \log(2\pi e(\det(\Sigma + \mathbb{E}[\sigma_P^2]I_d))^{\frac{1}{d}}) \right) - D_{KL}(P'||P_\theta)$$

$$= \frac{d}{2} \left( \log \left( \frac{e^{\frac{2}{d}\mathbb{H}(Y)} + 2\pi e(e^{\mathbb{E}[\log \sigma_Q^2]})}{2\pi e(\det(\Sigma + \mathbb{E}[\sigma_P^2]I_d))^{\frac{1}{d}}} \right) \right) - D_{KL}(P'||P_\theta)$$

$$= \frac{d}{2} \left( \log \left( \frac{e^{\frac{2}{d}\mathbb{H}(Y)} + 2\pi e(e^{\mathbb{E}[\log \sigma_Q^2]})}{2\pi e(\Pi_{i=1}^d(\lambda_i + \mathbb{E}[\sigma_P^2]))^{\frac{1}{d}}} \right) \right) - D_{KL}(P'||P_\theta)$$

(18)

□

## B  Details of Section 5

This section provides detailed information about the experimental implementation in Section 5.

For the density estimation model used in all comparison methods except LID, we employed ResFlow and utilized the library implemented by Stimper et al. (2023). As the optimizer, we used Adam (Diederik P. Kingma, 2015) with a weight decay of 1e-4 and a batch size of 64, and the initial learning rate was set to 1e-4. For the learning rate scheduler, we adopted CosineAnnealingWarmRestarts (Loshchilov & Hutter, 2017), setting the minimum learning rate to 1e-6. The cycle of decaying and increasing the learning rate was initially configured to half of the total number of epochs. However, since this setting caused training instability, we modified it by setting the entire number of epochs as a single cycle. In the ResFlow architecture, the number of multiscale blocks was fixed at three. For CIFAR-10/100, SVHN, and CelebA, the latent dimensionality was set to 128 and each multiscale block comprised 12 layers, while for MNIST and FashionMNIST the latent dimensionality and the number of layers per block were set to 64 and 5, respectively. In addition, the number of training epochs was set to 100 for CIFAR-10/100. The rectification threshold of ReAct $\beta$ was determined by randomly sampling 1,000 in-distribution training embedding vectors and setting the cutoff to their 90th percentile value, following Sun et al. (2021), without requiring access to OOD test data. Additionally, we do not apply pixel-value clipping after perturbation. Since the perturbation scale is kept sufficiently small, the perturbed inputs remain close to the original pixel range and do not introduce numerical instability in the flow model, as verified by the hyperparameter sensitivity analysis in Figure 4.

When calculating the likelihood of input data, we calculated the log-likelihood using the same density model across all implemented methods. For the complexity-based method, we employed the PNG compressor available in the Python OpenCV library (Bradski, 2000) to compute the number of bits, which was then used to perform the complexity calculation. In the likelihood ratio method, the background model was trained with the same hyperparameters as those used for estimating the density of the original input data, while the hyperparameter $\alpha$, which controls the probability of pixel perturbation, was set to 0.2. For the method utilizing Gaussian Mixture Models (GMMs), we implemented the model using scikit-learn (Pedregosa et al., 2011), with the number of mixture components fixed at 3, consistent with the original paper. Pytorch 2.4.1 (Paszke et al., 2019) was used to implement the density estimation model, and the computing environment used in the experiment was AMD Ryzen 9 7950X for CPU and RTX 4090Ti for GPU.

## C  Ablation Study of SPEM

### C.1  Meaning of Likelihood

Building upon the empirical success of SPEM and the entropy-based theoretical framework, a natural question arises: *what does model likelihood measure in practice?* As shown in Section 5, our results with SPEM indicate that regions assigned high likelihood are often those where a lower-entropy distribution dominates the comparison, rather than regions defined by semantic similarity. Thus, the likelihood value itself does not encode semantics; instead, for a given input it tends to reflect how concentrated the underlying distribution is. This observation is consistent with the phenomenon of likelihood inversion, whereby an OOD—despite being semantically distinct from the ID—can attain higher likelihood values when its entropy is sufficiently lower.

### C.2  Sensitivity of Likelihood

To examine the performance trend with respect to changes in the entropy of the OOD , we conducted the experiment illustrated in Figure 3. After training Glow on a real image dataset, we set the OOD to a Gaussian $\mathcal{N}(0, \sigma^2 I_d)$ and evaluated AUROC using only log-likelihood for detection as $\sigma$ varied. Unlike the SPEM framework, which injects noise into OOD samples composed of real images, this experiment directly manipulates the entropy of an OOD consisting purely of noise. As shown in Figure 3, the AUROC for SVHN rises steeply at small values of $\sigma$, reaching high performance earlier than CIFAR-10 and CIFAR-100.

Interestingly, CelebA follows a pattern much closer to SVHN than to CIFAR-10: its AUROC also escalates at similar noise levels, despite CelebA having entropy comparable to CIFAR-10. This discrepancy indicates that the observed sensitivity cannot be explained by entropy differences alone. While the convergence of AUROC toward 1 as $\sigma$ increases across all datasets suggests that relative entropy differences strongly affect likelihood ordering, this phenomenon can be further understood through the role of the KL-divergence $D_{KL}(Q\|P_\theta)$ in Equation 1. While our theoretical analysis establishes lower bounds on the expected log-likelihood gap, the $D_{\mathrm{KL}}(Q\|P_\theta)$ remains uncontrolled, meaning that the exact likelihood expectation gap is not directly guaranteed. In future work, we expect that a likelihood-based detection methodology that accounts for not only the entropic aspect of the ID but also the KL-divergence term $D_{\mathrm{KL}}(Q\|P_\theta)$ will yield more reliable detection performance with density estimation models. Since computing $D_{\mathrm{KL}}(Q\|P_\theta)$ is generally infeasible in practice, such approaches would need to exploit prior knowledge about the specific cases where likelihood-based methods fail as inductive biases or auxiliary signals to guide the generative model's design.

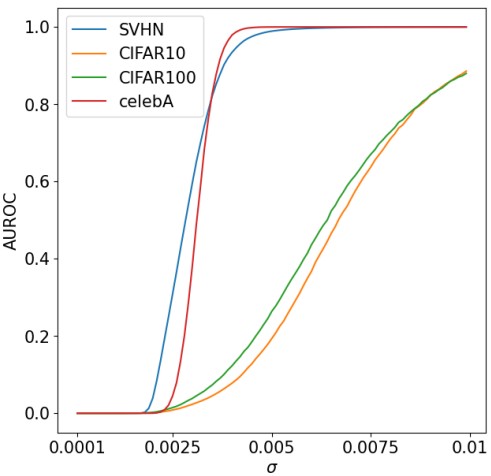

Figure 3: AUROC for IDs composed of real images as a function of $\sigma$ when the OOD is $\mathcal{N}(0, \sigma^2 I_d)$.

## C.3 Hyperparameter Sensitivity

In SPEM, we conducted an experiment to examine how adjusting the global entropy manipulation intensity $\alpha$ affects OOD detection performance, as reported in Figure 4. According to Figure 4, when $\alpha$ is small, the entropy gap between ID/OOD samples is not sufficiently enlarged, resulting in relatively low AUROC values. However, as $\alpha$ increases, the AUROC steadily improves and eventually reaches a plateau. This experiment demonstrates that by setting $\alpha$ to a sufficiently large value (e.g., $\alpha = 0.4$), the performance becomes robust to small variations around this range while achieving high AUROC. Furthermore, we observed that excessively large values of $\alpha$ introduce numerical stability issues in density estimation. Therefore, we recommend setting $\alpha$ to an adequately large but not excessive value, since additional performance gains beyond a certain threshold are limited. Importantly, $\alpha$ can be selected without access to the OOD test dataset; a held-out dataset disjoint from the test OOD can serve as a validation set. Based on the plateau observed within $\alpha \in [0.3, 0.5]$, $\alpha = 0.4$ can be set as a representative value.

## C.4 Effect of Feature Extractor

Table 5: OOD detection performance using ResFlow according to changes in feature extractor.

| $P$ (In) | CIFAR-10 | CIFAR-100 | CelebA | CIFAR-10 | FashionMNIST | SVHN | SVHN | SVHN | CelebA | MNIST |
| $Q$ (Out) | SVHN | SVHN | SVHN | CelebA | MNIST | CIFAR-10 | CIFAR-100 | CelebA | CIFAR-10 | FashionMNIST |
|---|---|---|---|---|---|---|---|---|---|---|
| ResNet-50 | 0.9839 | 0.9435 | 0.9999 | 0.9547 | **0.9643** | 0.9991 | 0.9984 | 0.9995 | 0.9997 | 1.0000 |
| ResNet-101 | 0.9894 | **0.9536** | **1.0000** | 0.9691 | 0.8623 | 0.9996 | **0.9992** | 0.9999 | 0.9999 | **0.9999** |
| ResNet-152 | **0.9913** | 0.9359 | **1.0000** | **0.9830** | 0.9207 | **0.9997** | **0.9992** | 1.0000 | 0.9999 | 1.0000 |

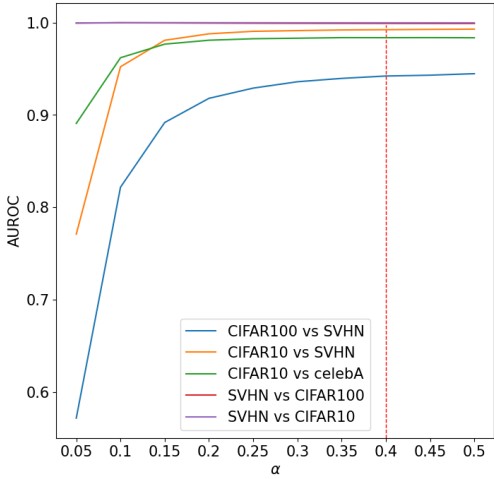

Figure 4: OOD detection performance according to $\alpha$, which controls the intensity of entropy manipulation. The legend indicates the experimental dataset pairs, with the former indicating in-distribution and the latter indicating OOD. All experiments were set up identically to those in Table 1.

Table 6: OOD detection performance using Glow according to changes in feature extractor.

| $P$ (In) | CIFAR-10 | CIFAR-100 | CelebA | CIFAR-10 | FashionMNIST | SVHN | SVHN | SVHN | CelebA | MNIST |
|---|---|---|---|---|---|---|---|---|---|---|
| $Q$ (Out) | SVHN | SVHN | SVHN | CelebA | MNIST | CIFAR-10 | CIFAR-100 | CelebA | CIFAR-10 | FashionMNIST |
| ResNet-50 | 0.9583 | 0.8712 | 0.9634 | 0.9498 | **0.9774** | 0.9965 | 0.9959 | 0.9975 | 0.9628 | **1.0000** |
| ResNet-101 | 0.9722 | **0.8897** | **0.9705** | 0.9621 | 0.8887 | 0.9979 | **0.9977** | 0.9992 | 0.9676 | 0.9999 |
| ResNet-152 | **0.9768** | 0.8655 | 0.9697 | **0.9782** | 0.9424 | **0.9985** | 0.9975 | **0.9993** | **0.9701** | **1.0000** |

We conducted experiments on the trend of OOD detection performance by changing the feature extractor that determines the similarity value $\lambda$ with embedding vector in memory bank, which controls the strength of entropy manipulation. As feature extractors, we used ResNet-50/101/152 pretrained on ImageNet, and the results are reported in Table 5 and 6. According to Table 5 and 6, ResNet-152, which has greater expressive power in our ablation study, generally achieved higher performance and thus we adopted it as the feature extractor in our main experiments. This can be interpreted as follows: the stronger the expressive power, the more pronounced the difference in similarity between ID/OOD, leading to the results shown in experiments. Based on this finding, one way to further improve the proposed SPEM is to fine-tune a model pretrained on the in-distribution dataset and then plug the tuned model into SPEM. Although this approach requires additional computational resources, it could yield even better OOD detection performance.

## C.5 Effect of ReAct

Table 7: Performance difference in OOD detection using ResFlow with and without the ReAct module.

| $P$ (In) | CIFAR-10 | CIFAR-100 | CelebA | CIFAR-10 | FashionMNIST | SVHN | SVHN | SVHN | CelebA | MNIST |
|---|---|---|---|---|---|---|---|---|---|---|
| $Q$ (Out) | SVHN | SVHN | SVHN | CelebA | MNIST | CIFAR-10 | CIFAR-100 | CelebA | CIFAR-10 | FashionMNIST |
| w/ ReAct | **0.9913** | **0.9359** | **1.0000** | 0.9830 | **0.9207** | **0.9997** | **0.9992** | **1.0000** | 0.9999 | **1.0000** |
| w/o ReAct | 0.9536 | 0.8471 | 0.9999 | **0.9839** | 0.9042 | 0.9994 | 0.9988 | 0.9999 | **0.9999** | **1.0000** |

Table 8: Performance difference in OOD detection using Glow with and without the ReAct module.

| $P$ (In) | CIFAR-10 | CIFAR-100 | CelebA | CIFAR-10 | FashionMNIST | SVHN | SVHN | SVHN | CelebA | MNIST |
|---|---|---|---|---|---|---|---|---|---|---|
| $Q$ (Out) | SVHN | SVHN | SVHN | CelebA | MNIST | CIFAR-10 | CIFAR-100 | CelebA | CIFAR-10 | FashionMNIST |
| w/ ReAct | **0.9768** | **0.8655** | 0.9697 | 0.9782 | **0.9424** | 0.9985 | 0.9975 | 0.9993 | 0.9701 | **1.0000** |
| w/o ReAct | 0.9116 | 0.7570 | **0.9806** | **0.9795** | 0.9284 | **0.9990** | **0.9983** | **0.9998** | **0.9886** | **1.0000** |

We incorporated ReAct, a method that rectifies the activations of the penultimate layer in pretrained models, into our proposed framework SPEM to enhance OOD detection performance. The motivation behind this integration is that we expected that the similarity gap between test vectors sampled from ID/OOD and the in-distribution training vectors stored in the memory bank would be amplified, thereby improving detection performance. To validate this hypothesis, we measured the performance of SPEM with and without the ReAct module, and the results are reported in Table 7 and 8. Table 7 demonstrates that incorporating ReAct into SPEM with ResFlow consistently improves performance across the majority of dataset pairs. In contrast, Table 8 shows that for dataset pairs where likelihood-based methods already perform well, the use of ReAct within SPEM results in only a marginal decrease in performance. Nevertheless, in scenarios where likelihood-based approaches fail, ReAct provides substantial improvements, with instances of performance degradation remaining relatively minor. This observation suggests that ReAct is particularly beneficial in hard dataset pairs (i.e., when the entropy of the in-distribution exceeds that of the OOD distribution). Consistent with the findings reported in the original ReAct paper, these results provide evidence that rectification amplifies the separation between in-distribution and OOD embedding vectors, thereby highlighting the difference in similarity and ultimately improving detection performance.

## C.6  Performance Comparison of SPEM using Glow

To examine the compatibility and performance consistency of SPEM with different density estimation models, we additionally employed Glow to estimate the in-distribution and compared its performance. Most experimental settings were aligned with those of ResFlow, with the exception that for CIFAR-10/100, SVHN, and CelebA, the latent dimensionality was set to 256 and each multiscale block comprised 16 layers, while for MNIST and FashionMNIST the latent dimensionality and the number of layers per block were set to 64 and 5, respectively. The number of training epochs was fixed at 300 for CIFAR-10/100, SVHN, and CelebA, and 80 for MNIST and FashionMNIST. For SPEM, $\alpha$ was set to 0.3, and for SPEM-noise it was set to 0.1. The experimental results are reported in Table 9.

Table 9: OOD detection performance on real image dataset using Glow. The five dataset pairs on the left are widely recognized as cases where likelihood-based OOD detection fails, and the remaining pairs do not exhibit problematic behavior in the estimated likelihood. For each dataset pair, the values that ranked within the top two across all compared models are highlighted in bold.

| $P$ (In) | CIFAR-10 | CIFAR-100 | CelebA | CIFAR-10 | FashionMNIST | SVHN | SVHN | SVHN | CelebA | MNIST |
| $Q$ (Out) | SVHN | SVHN | SVHN | CelebA | MNIST | CIFAR-10 | CIFAR-100 | CelebA | CIFAR-10 | FashionMNIST |
|---|---|---|---|---|---|---|---|---|---|---|
| Likelihood | 0.0807 | 0.0939 | 0.1292 | 0.5144 | 0.7359 | 0.9919 | 0.9906 | 0.9991 | 0.7408 | 0.9997 |
| Complexity | 0.8718 | 0.8319 | 0.9737 | 0.5566 | 0.8604 | 0.5270 | 0.6020 | 0.6163 | 0.7564 | 0.9712 |
| Typicality ($\sqrt{d}$) | 0.6530 | 0.6764 | **0.9998** | 0.6297 | 0.7365 | 0.9783 | 0.9744 | 0.9976 | 0.9253 | 0.9997 |
| Typicality (Entropy) | 0.4929 | 0.4762 | 0.7300 | 0.4283 | 0.5878 | 0.9894 | 0.9876 | 0.9990 | 0.7237 | 0.9997 |
| Likelihood Ratio | 0.8477 | 0.6020 | 0.8550 | 0.7334 | **0.9616** | 0.1078 | 0.1398 | 0.1548 | 0.5146 | 0.9875 |
| GMM | 0.8744 | 0.6658 | **0.9998** | 0.6174 | 0.7964 | 0.9762 | 0.9751 | 0.9976 | 0.9336 | 0.9996 |
| LID | 0.6573 | **0.9330** | 0.9490 | 0.6550 | **0.9510** | 0.9790 | 0.9860 | 0.9960 | 0.9390 | **1.0000** |
| SPEM | **0.9768** | 0.8655 | 0.9697 | **0.9782** | 0.9424 | **0.9985** | **0.9975** | **0.9993** | **0.9701** | **1.0000** |
| SPEM-noise | **0.9916** | **0.9343** | **0.9999** | **0.9830** | 0.9424 | **0.9996** | **0.9988** | **0.9999** | **0.9993** | **1.0000** |

As shown in Table 9, SPEM with Glow consistently outperforms other comparison methods across most ID/OOD pairs. Methods that rely solely on likelihood yield extremely low AUROC when the in-distribution has higher entropy than the OOD distribution (e.g., when SVHN is used as the in-distribution). In these cases, both the comparison methods and our proposed approach clearly surpass the likelihood-only baseline. Conversely, when the in-distribution entropy is lower than that of the OOD, some comparison methods perform worse than the likelihood baseline. Nevertheless, our approach consistently outperforms all comparison methods, even in these challenging scenarios. In addition, we observed that the performance gap between SPEM and SPEM-noise was larger when using Glow compared to ResFlow. This result suggests that the probability models estimated by different density estimators are inherently distinct. Although SPEM using Glow achieved slightly lower performance than when using ResFlow, the fact that it still outperformed most comparison models demonstrates that SPEM's superiority in likelihood-based OOD detection is largely agnostic to the choice of the underlying density model.

### C.7 Analysis of SPEM-noise

In the OOD detection results reported in Tables 1 and 9 SPEM-noise performs comparably to and in several cases outperforms SPEM. SPEM-noise disregards the original test data and instead measures the log-likelihood by feeding into the density estimation model only Gaussian noise, the magnitude of which is determined by the maximum similarity between the test vector and the in-distribution embedding vectors stored in the memory bank. Although this input contains no substantive semantic information, SPEM-noise nevertheless achieves performance comparable to or even superior to SPEM, which may appear counter-intuitive. We account for this phenomenon by analyzing the increment in the expected log-likelihoods of ID/OOD samples under SPEM-noise and SPEM. Let the in-distribution be denoted by $X \sim P$, OOD by $Y \sim Q$ that has covariance matrix $\Sigma_Q$, the noise distribution applied to the in-distribution under SPEM by $Z \sim \mathcal{N}(0, \sigma_P^2 I_d)$, and the noise distribution applied to the OOD by $Z' \sim \mathcal{N}(0, \sigma_Q^2 I_d)$. Also, we denote the density estimation model by $P_\theta$ and the manipulated ID/OOD induced by the noise be denoted as $X + Z \sim P'$ and $Y + Z' \sim Q'$, respectively. In the implementation of SPEM, the noise is constructed to depend on similarity, so that $\sigma_P^2$ and $\sigma_Q^2$ in practice follow specific distributions. However, for analytical tractability, we set them as constants in our analysis in this section. Then, the difference in the expected log-likelihoods computed under SPEM can be derived as follows:

$$
\begin{aligned}
&\mathbb{E}_{\mathbf{x} \sim P'}[\log P_\theta(\mathbf{x})] - \mathbb{E}_{\mathbf{x} \sim Q'}[\log P_\theta(\mathbf{x})] \\
&= D_{KL}(Q' || P_\theta) - D_{KL}(P' || P_\theta) + \mathbb{H}(Q') - \mathbb{H}(P') \\
&= D_{KL}(Y + Z' || P_\theta) - D_{KL}(X + Z || P_\theta) + \mathbb{H}(Y + Z') - \mathbb{H}(X + Z)
\end{aligned}
$$

In addition, for the case of SPEM-noise, the difference in the expected log-likelihoods can be derived as follows:

$$
\begin{aligned}
&\mathbb{E}_{\mathbf{x} \sim Z}[\log P_\theta(\mathbf{x})] - \mathbb{E}_{\mathbf{x} \sim Z'}[\log P_\theta(\mathbf{x})] \\
&= D_{KL}(Z' || P_\theta) - D_{KL}(Z || P_\theta) + \mathbb{H}(Z') - \mathbb{H}(Z)
\end{aligned}
$$

Therefore, under SPEM-noise, the log-likelihood difference increment relative to that of SPEM denoted by $\Delta$, can be derived as follows:

$$
\begin{aligned}
\Delta &= \mathbb{E}_{\mathbf{x} \sim Z}[\log P_\theta(\mathbf{x})] - \mathbb{E}_{\mathbf{x} \sim Z'}[\log P_\theta(\mathbf{x})] - (\mathbb{E}_{\mathbf{x} \sim P'}[\log P_\theta(\mathbf{x})] - \mathbb{E}_{\mathbf{x} \sim Q'}[\log P_\theta(\mathbf{x})]) \\
&= D_{KL}(Z' || P_\theta) - D_{KL}(Z || P_\theta) + \mathbb{H}(Z') - \mathbb{H}(Z) \\
&\quad - (D_{KL}(Y + Z' || P_\theta) - D_{KL}(X + Z || P_\theta) + \mathbb{H}(Y + Z') - \mathbb{H}(X + Z)) \\
&= D_{KL}(Z' || P_\theta) - D_{KL}(Z || P_\theta) - (D_{KL}(Y + Z' || P_\theta) - D_{KL}(X + Z || P_\theta)) \\
&\quad + \mathbb{H}(Z') - \mathbb{H}(Z) - (\mathbb{H}(Y + Z') - \mathbb{H}(X + Z))
\end{aligned}
$$

We decompose this into the increment of the entropy term denoted by $\Delta_E$, and the increment of the KL-divergence denoted by $\Delta_{KL}$, and express it using the following notation:

$$
\begin{aligned}
\Delta_{KL} &= D_{KL}(Z' || P_\theta) - D_{KL}(Z || P_\theta) - (D_{KL}(Y + Z' || P_\theta) - D_{KL}(X + Z || P_\theta)) \\
\Delta_E &= \mathbb{H}(Z') - \mathbb{H}(Z) - (\mathbb{H}(Y + Z') - \mathbb{H}(X + Z))
\end{aligned}
$$

To analyze these increments, we first examine the conditions under which $\Delta_E$ becomes positive, so we derive Theorem C.1.

**Theorem C.1.** *Let $P$, $P_\theta$, and $Q$ be $d$-dimensional continuous probability distributions on $\mathbb{R}^d$. Let*

$$
X \sim P, \quad Y \sim Q, \quad Z \sim \mathcal{N}(0, \sigma_P^2 I_d), \quad Z' \sim \mathcal{N}(0, \sigma_Q^2 I_d),
$$

*and define*

$$X + Z \sim P', \quad Y + Z' \sim Q'.$$

*Let $Q$ have covariance matrix $\Sigma_Q$. Then, the sufficient condition of $\Delta_E > 0$ is*

$$\frac{\sigma_Q^2}{\sigma_P^2} > \frac{2\pi e\,(tr(\Sigma_Q))}{d e^{\frac{2}{d}\mathbb{H}(X)}}.$$

*Proof.* By applying the entropy power inequality and the maximum-entropy property of the Gaussian distribution, both of which are used in the proof of Theorem 4.1, we can derive the following results:

$$
\begin{aligned}
\mathbb{H}(Z') - \mathbb{H}(Z) &= \frac{d}{2}\log\frac{\sigma_Q^2}{\sigma_P^2} \\
\mathbb{H}(Y + Z') &\le \frac{d}{2}\log(2\pi e(\det(\Sigma_Q + \sigma_Q^2 I_d))^{\frac{1}{d}}) \\
\mathbb{H}(X + Z) &\ge \frac{d}{2}\log(e^{\frac{2}{d}\mathbb{H}(X)} + 2\pi e\sigma_P^2)
\end{aligned}
\tag{19}
$$

Therefore, we can derive as follows:

$$
\begin{aligned}
\Delta_E &= \mathbb{H}(Z') - \mathbb{H}(Z) - (\mathbb{H}(Y + Z') - \mathbb{H}(X + Z)) \\
&= \frac{d}{2}\log\frac{\sigma_Q^2}{\sigma_P^2} - (\mathbb{H}(Y + Z') - \mathbb{H}(X + Z)) \\
&\ge \frac{d}{2}\log\frac{\sigma_Q^2}{\sigma_P^2} - \frac{d}{2}\log(2\pi e(\det(\Sigma_Q + \sigma_Q^2 I_d))^{\frac{1}{d}}) + \frac{d}{2}\log(e^{\frac{2}{d}\mathbb{H}(X)} + 2\pi e\sigma_P^2) \\
&= \frac{d}{2}\log\frac{\sigma_Q^2}{\sigma_P^2} - \frac{d}{2}\left(\log\frac{2\pi e(\det(\Sigma_Q + \sigma_Q^2 I_d))^{\frac{1}{d}}}{e^{\frac{2}{d}\mathbb{H}(X)} + 2\pi e\sigma_P^2}\right)
\end{aligned}
\tag{20}
$$

Finally, we derive the following sufficient condition for $\Delta_E > 0$:

$$
\begin{aligned}
\Delta_E &= \frac{d}{2}\log\frac{\sigma_Q^2}{\sigma_P^2} - \frac{d}{2}\left(\log\frac{2\pi e(\det(\Sigma_Q + \sigma_Q^2 I_d))^{\frac{1}{d}}}{e^{\frac{2}{d}\mathbb{H}(X)} + 2\pi e\sigma_P^2}\right) > 0 \\
&\Rightarrow \frac{d}{2}\log\frac{\sigma_Q^2}{\sigma_P^2} > \frac{d}{2}\left(\log\frac{2\pi e(\det(\Sigma_Q + \sigma_Q^2 I_d))^{\frac{1}{d}}}{e^{\frac{2}{d}\mathbb{H}(X)} + 2\pi e\sigma_P^2}\right) \\
&\Rightarrow \frac{\sigma_Q^2}{\sigma_P^2} > \frac{2\pi e(\det(\Sigma_Q + \sigma_Q^2 I_d))^{\frac{1}{d}}}{e^{\frac{2}{d}\mathbb{H}(X)} + 2\pi e\sigma_P^2}
\end{aligned}
\tag{21}
$$

A more conservative variance ratio inequality can be derived by eliminating both $\sigma_P^2$ and $\sigma_Q^2$ from the RHS of the inequality, first we derive upper bound of determinant using AM-GM inequality:

$$
\begin{aligned}
(\det(\Sigma_Q + \sigma_Q^2 I_d))^{\frac{1}{d}} &= (\Pi_{i=1}^{d}(\lambda_i + \sigma_Q^2))^{\frac{1}{d}} \\
&\le \frac{1}{d}(\Sigma_{i=1}^{d}(\lambda_i + \sigma_Q^2)) \\
&= \frac{\mathrm{tr}(\Sigma_Q)}{d} + \sigma_Q^2
\end{aligned}
\tag{22}
$$

where $\lambda_i$ is $i$-th eigenvalue of $\Sigma_Q$. Then, we derive the following conservative sufficient condition for $\Delta_E > 0$:

$$\frac{\sigma_Q^2}{\sigma_P^2} > \frac{2\pi e(\frac{\text{tr}(\Sigma_Q)}{d} + \sigma_Q^2)}{e^{\frac{2}{d}\mathbb{H}(X)} + 2\pi e\sigma_P^2}$$

$$\Rightarrow \sigma_Q^2 e^{\frac{2}{d}\mathbb{H}(X)} + \sigma_P^2\sigma_Q^2 2\pi e > \sigma_P^2 2\pi e\left(\frac{\text{tr}(\Sigma_Q)}{d}\right) + \sigma_P^2\sigma_Q^2 2\pi e$$

$$\Rightarrow \sigma_Q^2 e^{\frac{2}{d}\mathbb{H}(X)} > \sigma_P^2 2\pi e\left(\frac{\text{tr}(\Sigma_Q)}{d}\right) \tag{23}$$

$$\Rightarrow \frac{\sigma_Q^2}{\sigma_P^2} > \frac{2\pi e\left(\text{tr}(\Sigma_Q)\right)}{de^{\frac{2}{d}\mathbb{H}(X)}}$$

$\square$

This implies that, regardless of the entropy gap between the in-distribution and the out-of-distribution, it is always possible to find a ratio of $\sigma_Q^2$ to $\sigma_P^2$ that ensures $\Delta_E > 0$. Additionally, Theorem C.4 shows that $\Delta_E$ is monotonically increasing in $\sigma_Q^2$, and it is monotonically decreasing in $\sigma_P^2$ with Fisher information existence assumption.

**Lemma C.2** (De Bruijn's Identity, Guo et al. (2005)). *Let $X \in \mathbb{R}^d$ be a random vector with a well-defined density, and let $Z \sim \mathcal{N}(0, I_d)$ be an independent standard Gaussian random vector. For $t > 0$, define*

$$X_t = X + \sqrt{t}Z.$$

*Then the entropy of $X_t$ satisfies*

$$\frac{d}{dt}\mathbb{H}(X_t) = \frac{1}{2}\text{tr}(\boldsymbol{J}(X_t)),$$

*where*

$$\boldsymbol{J}(X_t) = \mathbb{E}\left[\nabla \log f_{X_t}(X_t) \nabla \log f_{X_t}(X_t)^\top\right]$$

*is the Fisher information matrix of $X_t$. At $t = 0$, the identity holds only under the assumption that $\boldsymbol{J}(X)$ exists.*

**Lemma C.3** (Fisher Information Inequality, Rioul (2010)). *Let $X$ and $Y$ be independent random vectors in $\mathbb{R}^d$ with non-zero finite Fisher information $J(X)$ and $J(Y)$. Then the Fisher information of their sum satisfies*

$$J(X + Y)^{-1} \geq J(X)^{-1} + J(Y)^{-1}.$$

**Theorem C.4.** *Let $P$, $P_\theta$, and $Q$ be $d$-dimensional continuous probability distributions on $\mathbb{R}^d$. Let*

$$X \sim P, \quad Y \sim Q, \quad Z \sim \mathcal{N}(0, \sigma_P^2 I_d), \quad Z' \sim \mathcal{N}(0, \sigma_Q^2 I_d),$$

*and define*

$$X + Z \sim P', \quad Y + Z' \sim Q'.$$

*Assume the Fisher information of $X$ and $Y$ are non-zero finite. Then, $\frac{\partial \Delta_E}{\partial \sigma_P^2} < 0$ and $\frac{\partial \Delta_E}{\partial \sigma_Q^2} > 0$.*

*Proof.* Let $X_{\sigma_P^2} = X + \sigma_P Z_1 = X + Z$ and $Y_{\sigma_Q^2} = Y + \sigma_Q Z_2 = Y + Z'$ such that $Z_1, Z_2 \sim \mathcal{N}(0, I_d)$. By Lemma C.2, we can derive as follows:

$$\frac{d}{d\sigma_P^2}\mathbb{H}(X + Z) = \frac{1}{2}\text{tr}(\mathbf{J}(X + Z)) = \frac{1}{2}J(X + Z)$$

$$\frac{d}{d\sigma_Q^2}\mathbb{H}(Y + Z') = \frac{1}{2}\text{tr}(\mathbf{J}(Y + Z')) = \frac{1}{2}J(Y + Z') \tag{24}$$

where $\mathbf{J}(\cdot)$ is Fisher information matrix.

Then, we can derive as follows using Lemma C.3:

$$
\begin{aligned}
J(X + Z)^{-1} &\geq J(X)^{-1} + J(Z)^{-1} = \frac{1}{J(Z)}\left(\frac{J(X) + J(Z)}{J(X)}\right) = \frac{\sigma_P^2}{d}\left(\frac{J(X) + J(Z)}{J(X)}\right) \\
J(Y + Z')^{-1} &\geq J(Y)^{-1} + J(Z')^{-1} = \frac{1}{J(Z')}\left(\frac{J(Y) + J(Z')}{J(Y)}\right) = \frac{\sigma_Q^2}{d}\left(\frac{J(Y) + J(Z')}{J(Y)}\right) \\
\therefore J(X + Z) &\leq \frac{d}{\sigma_P^2}\left(\frac{J(X)}{J(X) + J(Z)}\right), \quad J(Y + Z') \leq \frac{d}{\sigma_Q^2}\left(\frac{J(Y)}{J(Y) + J(Z')}\right)
\end{aligned}
\tag{25}
$$

By differentiating $\Delta_E$ with respect to $\sigma_P^2$, we obtain:

$$
\begin{aligned}
\frac{\partial \Delta_E}{\partial \sigma_P^2} &= \frac{\partial\left(\frac{d}{2}\log\frac{\sigma_Q^2}{\sigma_P^2} - (\mathbb{H}(Y + Z') - \mathbb{H}(X + Z))\right)}{\partial \sigma_P^2} \\
&= -\frac{d}{2\sigma_P^2} + \frac{\partial(\mathbb{H}(X + Z))}{\partial \sigma_P^2} \quad \left(\because \frac{\partial(\mathbb{H}(Y + Z'))}{\partial \sigma_P^2} = 0\right) \\
&= -\frac{d}{2\sigma_P^2} + \frac{J(X + Z)}{2} \quad (\because \text{Equation 24}) \\
&\leq -\frac{d}{2\sigma_P^2} + \frac{d}{2\sigma_P^2}\left(\frac{J(X)}{J(X) + J(Z)}\right) \quad (\because \text{Equation 25}) \\
&= -\frac{d}{2\sigma_P^2}\left(\frac{J(Z)}{J(X) + J(Z)}\right) \\
&< 0
\end{aligned}
\tag{26}
$$

Similarly, by differentiating $\Delta_E$ with respect to $\sigma_Q^2$, we obtain:

$$
\begin{aligned}
\frac{\partial \Delta_E}{\partial \sigma_Q^2} &= \frac{\partial\left(\frac{d}{2}\log\frac{\sigma_Q^2}{\sigma_P^2} - (\mathbb{H}(Y + Z') - \mathbb{H}(X + Z))\right)}{\partial \sigma_Q^2} \\
&= \frac{d}{2\sigma_Q^2} - \frac{\partial(\mathbb{H}(Y + Z'))}{\partial \sigma_Q^2} \quad \left(\because \frac{\partial(\mathbb{H}(X + Z))}{\partial \sigma_Q^2} = 0\right) \\
&= \frac{d}{2\sigma_Q^2} - \frac{J(Y + Z')}{2} \quad (\because \text{Equation 24}) \\
&\geq \frac{d}{2\sigma_Q^2} - \frac{d}{2\sigma_Q^2}\left(\frac{J(Y)}{J(Y) + J(Z')}\right) \quad (\because \text{Equation 25}) \\
&= \frac{d}{2\sigma_Q^2}\left(\frac{J(Z')}{J(Y) + J(Z')}\right) \\
&> 0
\end{aligned}
\tag{27}
$$

$\square$

Therefore, by Theorems C.1 and C.4, we establish that $\Delta_E$ not only becomes strictly positive once the noise variance ratio $\sigma_Q^2/\sigma_P^2$ exceeds a certain threshold, but also exhibits opposite monotonic behaviors with respect to the ID/OOD noise variances. Specifically, $\Delta_E$ exhibits a negative gradient with respect to $\sigma_P^2$ and a positive gradient with respect to $\sigma_Q^2$. Consequently, by decreasing $\sigma_P^2$ while simultaneously increasing $\sigma_Q^2$ such that their ratio surpasses the identified threshold, one can guarantee that $\Delta_E$, as a constituent component of $\Delta$, remains positive and strictly increases.

Since the overall increment $\Delta$ is influenced by $\Delta_{\mathrm{KL}}$, an analysis of $\Delta_{\mathrm{KL}}$ is required in order to explain the expected log-likelihood under both SPEM-noise and SPEM. However, unlike entropy, $\Delta_{\mathrm{KL}}$ is difficult to bound due to the arbitrariness of the OOD setting. Therefore, we first derive in Theorem C.11 a two-sided bound on $\Delta$ under the assumption that $\log P_\theta$ has the $L$-Lipschitz property, in order to analyze the bound of the overall difference $\Delta$.

**Definition C.5** (p-Wasserstein Distance). Let $P$, $Q$ be probability measures on $\mathbb{R}^d$. Then, p-Wasserstein distance between $P$ and $Q$ is

$$W_p(P,Q) := \inf_{\gamma \in \Pi(P,Q)} \left( \int ||x - y||^p d\gamma(x,y) \right)^{\frac{1}{p}}.$$

**Lemma C.6** (Wasserstein Distance Inequality, Santambrogio (2015)). *Let $P$, $Q$ be probability measures on $\mathbb{R}^d$ with finite p-th moments. Then,*

$$W_1(P,Q) \le W_p(P,Q), \ \ s.t \ \ p > 1.$$

**Lemma C.7** (Triangle Inequality of Wasserstein Distance, Santambrogio (2015)). *Let $P$, $Q$, $R$ be probability measures on $\mathbb{R}^d$ with finite p-th moments. Then,*

$$W_p(P,R) \le W_p(P,Q) + W_p(Q,R) \ \ for \ \ p \ge 1.$$

**Lemma C.8** (Wasserstein Distance between Gaussians, Givens & Shortt (1984)). *Let $Z_1 \sim \mathcal{N}(0, \sigma_1^2 I_d)$, $Z_2 \sim \mathcal{N}(0, \sigma_2^2 I_d)$. Then,*

$$W_2(Z_1, Z_2) = \sqrt{d}|\sigma_1 - \sigma_2|.$$

**Lemma C.9** (Convolution of Wasserstein Distance). *Let $P$, $Q$, $Z$ be probability measures on $\mathbb{R}^d$ with finite p-th moments. Then,*

$$W_p(P * Z, Q * Z) \le W_p(P,Q) \ \ for \ \ p \ge 1.$$

*Proof.* Let $\gamma^* \in \Pi(P,Q)$ is optimal coupling of $W_p(P,Q)$ and $\Pi(P,Q)$ is coupling of $P$ and $Q$. Then,

$$W_p(P,Q)^p = \int ||x - y||^p d\gamma^*(x,y) \tag{28}$$

Let $T : \mathbb{R}^d \times \mathbb{R}^d \times \mathbb{R}^d \to \mathbb{R}^d \times \mathbb{R}^d$ such that $T(\mathbf{x}, \mathbf{y}, \mathbf{z}) = (\mathbf{x} + \mathbf{z}, \mathbf{y} + \mathbf{z}) = (\mathbf{u}, \mathbf{v})$ and $\tilde{\gamma} := T_\#(\gamma^* \otimes Z)$ pushforward of $\gamma^* \otimes Z$ by $T$. Since $U \sim P * Z$ and $V \sim Q * Z$, it follows that $\tilde{\gamma} \in \Pi(P * Z, Q * Z)$. Consequently,

$$
\begin{aligned}
&\int ||\mathbf{u} - \mathbf{v}||^p d\tilde{\gamma} \\
&= \int ||(\mathbf{x} + \mathbf{z}) - (\mathbf{y} + \mathbf{z})||^p d(\gamma^* \otimes Z) \\
&= \int ||\mathbf{x} - \mathbf{y}||^p d\gamma^* \\
&= W_p(P,Q)^p
\end{aligned}
\tag{29}
$$

Taking the infimum on both sides, we can derive as follows:

$$\inf_{\tilde{\gamma} \in \Pi(P*Z, Q*Z)} \int ||\mathbf{u} - \mathbf{v}||^p d\tilde{\gamma} = W_p(P * Z, Q * Z)^p \le W_p(P,Q)^p \tag{30}$$

$\square$

**Lemma C.10** (*L-Lipschitz Wasserstein Distance Inequality*)**.** *Let $P$, $Q$ be probability measures on $\mathbb{R}^d$ with finite first moments. Let $f : \mathbb{R}^d \to \mathbb{R}$ be $L$-Lipschitz. Then,*

$$|\mathbb{E}_P[f] - \mathbb{E}_Q[f]| \leq LW_1(P, Q)$$

*Proof.* Because $f$ is $L$-Lipschitz, we can derive as follows:

$$
\begin{aligned}
|\mathbb{E}_P[f] - \mathbb{E}_Q[f]| &= \left| \int f(\mathbf{x})dP(\mathbf{x}) - \int f(\mathbf{y})dQ(\mathbf{y}) \right| \\
&= \left| \int f(\mathbf{x}) - f(\mathbf{y})d\gamma(\mathbf{x}, \mathbf{y}) \right| \\
&\leq \int |f(\mathbf{x}) - f(\mathbf{y})| \, d\gamma(\mathbf{x}, \mathbf{y}) \\
&\leq L \int ||\mathbf{x} - \mathbf{y}||d\gamma(\mathbf{x}, \mathbf{y})
\end{aligned}
\tag{31}
$$

where $\gamma$ is arbitrary coupling of $P$ and $Q$. Taking the infimum on both sides, we can derive as follows:

$$
\begin{aligned}
|\mathbb{E}_P[f] - \mathbb{E}_Q[f]| &\leq \inf_{\gamma \in \Pi(P,Q)} L \int ||\mathbf{x} - \mathbf{y}||d\gamma(\mathbf{x}, \mathbf{y}) \\
&= LW_1(P, Q)
\end{aligned}
\tag{32}
$$

$\square$

**Theorem C.11.** *Let $P$, $P_\theta$, and $Q$ be $d$-dimensional continuous probability distributions on $\mathbb{R}^d$ and have finite first and second moments. Let*

$$X \sim P, \quad Y \sim Q, \quad Z \sim \mathcal{N}(0, \sigma_P^2 I_d), \quad Z' \sim \mathcal{N}(0, \sigma_Q^2 I_d),$$

*and define*

$$X + Z \sim P', \quad Y + Z' \sim Q'.$$

*Let $f(\mathbf{x}) = \log P_\theta(\mathbf{x})$ be $L$-Lipschitz. Then*

$$|\Delta| \leq L(W_2(P, Q) + 2\sqrt{d}|\sigma_Q - \sigma_P|)$$

*Proof.* By definition of $\Delta$,

$$\Delta = \mathbb{E}_{\mathbf{x} \sim Z}[\log P_\theta(\mathbf{x})] - \mathbb{E}_{\mathbf{x} \sim Z'}[\log P_\theta(\mathbf{x})] - (\mathbb{E}_{\mathbf{x} \sim P'}[\log P_\theta(\mathbf{x})] - \mathbb{E}_{\mathbf{x} \sim Q'}[\log P_\theta(\mathbf{x})]) \tag{33}$$

Because of triangle inequality, we can derive as follows:

$$
\begin{aligned}
|\Delta| &= |\mathbb{E}_{\mathbf{x} \sim Z}[\log P_\theta(\mathbf{x})] - \mathbb{E}_{\mathbf{x} \sim Z'}[\log P_\theta(\mathbf{x})] - (\mathbb{E}_{\mathbf{x} \sim P'}[\log P_\theta(\mathbf{x})] - \mathbb{E}_{\mathbf{x} \sim Q'}[\log P_\theta(\mathbf{x})])| \\
&\leq |\mathbb{E}_{\mathbf{x} \sim Z}[\log P_\theta(\mathbf{x})] - \mathbb{E}_{\mathbf{x} \sim Z'}[\log P_\theta(\mathbf{x})]| + |\mathbb{E}_{\mathbf{x} \sim P'}[\log P_\theta(\mathbf{x})] - \mathbb{E}_{\mathbf{x} \sim Q'}[\log P_\theta(\mathbf{x})]|
\end{aligned}
\tag{34}
$$

From above equation, we obtain the following:

$$
\begin{aligned}
|\Delta| &\leq |\mathbb{E}_{\mathbf{x} \sim Z}[\log P_\theta(\mathbf{x})] - \mathbb{E}_{\mathbf{x} \sim Z'}[\log P_\theta(\mathbf{x})]| + |\mathbb{E}_{\mathbf{x} \sim P'}[\log P_\theta(\mathbf{x})] - \mathbb{E}_{\mathbf{x} \sim Q'}[\log P_\theta(\mathbf{x})]| \\
&\leq LW_1(Z, Z') + LW_1(P', Q') \ \ (\because \text{Lemma } C.10) \\
&\leq LW_1(Z, Z') + LW_1(P', Q * Z) + LW_1(Q * Z, Q')(\because \text{Lemma } C.7) \\
&\leq LW_1(Z, Z') + LW_1(P', Q * Z) + LW_1(Z, Z')(\because \text{Lemma } C.9) \\
&\leq 2LW_1(Z, Z') + LW_1(P, Q)(\because \text{Lemma } C.9) \\
&\leq 2LW_2(Z, Z') + LW_2(P, Q)(\because \text{Lemma } C.6) \\
&\leq L(2\sqrt{d}|\sigma_Q - \sigma_P| + W_2(P, Q))(\because \text{Lemma } C.8)
\end{aligned}
\tag{35}
$$

$\square$

By Theorem C.11, we have shown that when $\log P_\theta$ is $L$-Lipschitz, $\Delta$ admits a two-sided bound that consists of the Lipschitz constant, the Wasserstein distance between $P$ and $Q$, and the difference in noise intensity. Although the bound on $\Delta$ includes positive values, it does not explicitly provide conditions under which $\Delta$ becomes strictly positive. Moreover, for analytical convenience, we assumed that the log-likelihood is $L$-Lipschitz; however, this assumption deviates significantly from practical distributions. Therefore, we relax this condition and, in Theorem C.14, derive the bound under the more realistic assumptions that the negative log-likelihood is $L$-smooth and $\lambda$-semiconvex. These conditions can accommodate probability distributions that are weakly multi-modal and provide a more practical setting compared to the $L$-Lipschitz assumption.

**Definition C.12** ($\lambda$-semiconvex function, Payne & Redaelli (2023)). Let $f : \mathbb{R}^d \to \mathbb{R}$ be a differentiable function. We say that $f$ is $\lambda$-*semiconvex* for some $\lambda \geq 0$ if the function

$$\mathbf{x} \; \mapsto \; f(\mathbf{x}) + \frac{\lambda}{2}\|\mathbf{x}\|^2$$

is convex. Equivalently, for all $\mathbf{x}, \mathbf{y} \in \mathbb{R}^d$,

$$f(\mathbf{y}) \; \geq \; f(\mathbf{x}) + \langle \nabla f(\mathbf{x}), \mathbf{y} - \mathbf{x} \rangle - \frac{\lambda}{2}\|\mathbf{y} - \mathbf{x}\|^2.$$

**Definition C.13** ($L$-smooth function). Let $f : \mathbb{R}^d \to \mathbb{R}$ be a differentiable function. We say that $f$ is $L$-*smooth* for some $L > 0$ if, for all $\mathbf{x}, \mathbf{y} \in \mathbb{R}^d$,

$$\|\nabla f(\mathbf{x}) - \nabla f(\mathbf{y})\| \; \leq \; L\|\mathbf{x} - \mathbf{y}\|.$$

Equivalently, $f$ is $L$-smooth if and only if

$$f(\mathbf{y}) \; \leq \; f(\mathbf{x}) + \langle \nabla f(\mathbf{x}), \mathbf{y} - \mathbf{x} \rangle + \frac{L}{2}\|\mathbf{y} - \mathbf{x}\|^2, \quad \forall \mathbf{x}, \mathbf{y} \in \mathbb{R}^d.$$

**Theorem C.14.** *Let $P$, $P_\theta$, and $Q$ be $d$-dimensional continuous probability distributions on $\mathbb{R}^d$ and have finite first and second moments. Let*

$$X \sim P, \quad Y \sim Q, \quad Z \sim \mathcal{N}(0, \sigma_P^2 I_d), \quad Z' \sim \mathcal{N}(0, \sigma_Q^2 I_d),$$

*and define*

$$X + Z \sim P', \quad Y + Z' \sim Q'.$$

*Let $f(\mathbf{x}) = -\log P_\theta(\mathbf{x})$ be $\lambda$-semiconvex and $L$-smooth. Then*

$$\Delta \in [-C - \frac{d(\lambda + L)}{2}(\sigma_P^2 + \sigma_Q^2), -C + \frac{d(\lambda + L)}{2}(\sigma_P^2 + \sigma_Q^2)]$$

*where $C = \mathbb{E}_{\mathbf{x} \sim P}[\log P_\theta(\mathbf{x})] - \mathbb{E}_{\mathbf{x} \sim Q}[\log P_\theta(\mathbf{x})]$. Also, a sufficient condition for $\Delta > 0$ is $-\frac{2C}{d(\lambda + L)} > \sigma_P^2 + \sigma_Q^2$.*

*Proof.* Then, by definition of $\lambda$-semiconvex and $L$-smooth, we can derive as following:

$$f(\mathbf{x}) + \langle \nabla f(\mathbf{x}), \mathbf{y} - \mathbf{x} \rangle - \frac{\lambda}{2}\|\mathbf{y} - \mathbf{x}\|^2 \leq f(\mathbf{y}) \; \leq \; f(\mathbf{x}) + \langle \nabla f(\mathbf{x}), \mathbf{y} - \mathbf{x} \rangle + \frac{L}{2}\|\mathbf{y} - \mathbf{x}\|^2$$

$$\Rightarrow \langle \nabla f(\mathbf{x}), \mathbf{y} - \mathbf{x} \rangle - \frac{\lambda}{2}\|\mathbf{y} - \mathbf{x}\|^2 \leq f(\mathbf{y}) - f(\mathbf{x}) \leq \langle \nabla f(\mathbf{x}), \mathbf{y} - \mathbf{x} \rangle + \frac{L}{2}\|\mathbf{y} - \mathbf{x}\|^2 \tag{36}$$

$$\Rightarrow \langle \nabla f(\mathbf{0}), \mathbf{y} \rangle - \frac{\lambda}{2}\|\mathbf{y}\|^2 \leq f(\mathbf{y}) - f(\mathbf{0}) \leq \langle \nabla f(\mathbf{0}), \mathbf{y} \rangle + \frac{L}{2}\|\mathbf{y}\|^2$$

By plugging $Z$ into $\mathbf{y}$, we can derive the following:

$$\mathbb{E}[\langle \nabla f(\mathbf{0}), Z \rangle] - \frac{\lambda}{2}\mathbb{E}[\|Z\|^2] \le \mathbb{E}[f(Z)] - f(\mathbf{0}) \le \mathbb{E}[\langle \nabla f(\mathbf{0}), Z \rangle] + \frac{L}{2}\mathbb{E}[\|Z\|^2]$$
$$\Rightarrow -\frac{\lambda}{2}\mathbb{E}[\|Z\|^2] \le \mathbb{E}[f(Z)] - f(\mathbf{0}) \le +\frac{L}{2}\mathbb{E}[\|Z\|^2] \tag{37}$$

Then, by applying the same procedure to $Z'$ and expanding difference between the resulting inequalities, we obtain the following:

$$-\frac{\lambda}{2}\mathbb{E}[\|Z\|^2] - \frac{L}{2}\mathbb{E}[\|Z'\|^2] \le \mathbb{E}[f(Z)] - \mathbb{E}[f(Z')] \le +\frac{L}{2}\mathbb{E}[\|Z\|^2] + \frac{\lambda}{2}\mathbb{E}[\|Z'\|^2]$$
$$\Rightarrow -\frac{\lambda d}{2}\sigma_P^2 - \frac{Ld}{2}\sigma_Q^2 \le \mathbb{E}[f(Z)] - \mathbb{E}[f(Z')] \le +\frac{Ld}{2}\sigma_P^2 + \frac{\lambda d}{2}\sigma_Q^2$$
$$\Rightarrow -\frac{\lambda d}{2}\sigma_P^2 - \frac{Ld}{2}\sigma_Q^2 \le \mathbb{E}[-\log P_\theta(Z)] - \mathbb{E}[-\log P_\theta(Z')] \le +\frac{Ld}{2}\sigma_P^2 + \frac{\lambda d}{2}\sigma_Q^2 \tag{38}$$
$$\Rightarrow -\frac{Ld}{2}\sigma_P^2 - \frac{\lambda d}{2}\sigma_Q^2 \le \mathbb{E}[\log P_\theta(Z)] - \mathbb{E}[\log P_\theta(Z')] \le +\frac{\lambda d}{2}\sigma_P^2 + \frac{Ld}{2}\sigma_Q^2$$

Meanwhile, the log-likelihood expectations under $P'$ and $Q'$ can be expressed as follows:

$$\mathbb{E}_{\mathbf{x}\sim P'}[\log P_\theta(\mathbf{x})] = \mathbb{E}_{\mathbf{x}\sim P}[\log P_\theta(\mathbf{x})] + (\mathbb{E}_{\mathbf{x}\sim P'}[\log P_\theta(\mathbf{x})] - \mathbb{E}_{\mathbf{x}\sim P}[\log P_\theta(\mathbf{x})])$$
$$\mathbb{E}_{\mathbf{x}\sim Q'}[\log P_\theta(\mathbf{x})] = \mathbb{E}_{\mathbf{x}\sim Q}[\log P_\theta(\mathbf{x})] + (\mathbb{E}_{\mathbf{x}\sim Q'}[\log P_\theta(\mathbf{x})] - \mathbb{E}_{\mathbf{x}\sim Q}[\log P_\theta(\mathbf{x})]) \tag{39}$$

Then, by applying the same procedure to $\mathbb{E}_{\mathbf{x}\sim P'}[-\log P_\theta(\mathbf{x})] - \mathbb{E}_{\mathbf{x}\sim P}[-\log P_\theta(\mathbf{x})]$ as was done to obtain the two-sided bounds for log-likelihood expectation difference between $Z$ and $Z'$, we can derive the following:

$$-\frac{\lambda}{2}\mathbb{E}[\|Z\|^2] \le \mathbb{E}[f(X+Z)] - \mathbb{E}[f(X)] \le +\frac{L}{2}\mathbb{E}[\|Z\|^2]$$
$$\Rightarrow -\frac{\lambda d}{2}\sigma_P^2 \le \mathbb{E}[f(X+Z)] - \mathbb{E}[f(X)] \le +\frac{Ld}{2}\sigma_P^2 \tag{40}$$

Moreover, the following inequality holds for $\mathbb{E}_{\mathbf{x}\sim Q'}[-\log P_\theta(\mathbf{x})] - \mathbb{E}_{\mathbf{x}\sim Q}[-\log P_\theta(\mathbf{x})]$ as well

$$-\frac{\lambda d}{2}\sigma_Q^2 \le \mathbb{E}[f(Y+Z')] - \mathbb{E}[f(Y)] \le +\frac{Ld}{2}\sigma_Q^2 \tag{41}$$

Thus, we obtain the following two-sided bound:

$$-\frac{\lambda d}{2}\sigma_P^2 - \frac{Ld}{2}\sigma_Q^2 \le \mathbb{E}[f(X+Z)] - \mathbb{E}[f(X)] - (\mathbb{E}[f(Y+Z')] - \mathbb{E}[f(Y)]) \le +\frac{\lambda d}{2}\sigma_Q^2 + \frac{Ld}{2}\sigma_P^2 \tag{42}$$

Because Equation 42 contains a negative sign in the log-likelihood introduced by $f(\cdot)$, adding it to Equation 38 leads to the following inequality:

$$-C - \frac{d(\lambda + L)}{2}(\sigma_P^2 + \sigma_Q^2) \le \Delta \le -C + \frac{d(\lambda + L)}{2}(\sigma_P^2 + \sigma_Q^2) \tag{43}$$

where $C = \mathbb{E}_{\mathbf{x}\sim P}[\log P_\theta(\mathbf{x})] - \mathbb{E}_{\mathbf{x}\sim Q}[\log P_\theta(\mathbf{x})]$. Additionally, $-\frac{2C}{d(\lambda+L)} > \sigma_P^2 + \sigma_Q^2$ is sufficient to guarantee $\Delta > 0$.

$\square$

By Theorem C.14, we establish the existence of a guaranteed lower bound when the negative log-likelihood satisfies the $L$-smoothness and $\lambda$-semiconvexity conditions. This existence result holds when $C$ is negative,

that is, when the log-likelihood expectation of the OOD exceeds that of the in-distribution. This implies that there exist cases in which SPEM-noise, which does not directly incorporate the original data as input, attains a higher expected log-likelihood difference than SPEM. Consequently, this suggests that SPEM-noise may achieve superior OOD detection performance compared to SPEM in certain cases. For future work, it would be of interest to relax the assumption and investigate the effectiveness of SPEM-noise under weaker conditions, and to analyze the conditions under which $\Delta_{KL}$ becomes positive or takes values smaller than $\Delta_E$. Such analyses would further explain when and why SPEM-noise can yield improved OOD detection performance.

### C.8 Inference Time of SPEM

SPEM involves two auxiliary steps: extracting embedding vectors using a pretrained model and computing cosine similarity with the in-distribution embedding vectors stored in the memory bank. Let $d$ denote the feature dimension of the extractor and $n$ the number of embedding vectors in the memory bank. Then, the time complexity of this auxiliary process is $\mathcal{O}(nd)$, i.e., linear in both $n$ and $d$. For example, on CIFAR-10, the auxiliary process requires roughly 1 minute of computation about 60,000 train in-distribution images. After this step, the inference time is equivalent to that of methods relying solely on likelihood.

We also compared the computational cost of SPEM with other baselines:

- The likelihood ratio method requires training a separate background density model, resulting in a substantially higher computational burden.

- The typicality method has a similar cost to the likelihood-only method.

- The complexity method computes PNG-based bit lengths for each data point, which is lightweight.

- The GMM method incurs moderate cost due to the additional model fitting step in low-dimensional space.

- The LID method requires Jacobian computation and is therefore significantly more expensive, which is why we used the reported performance from the original paper.

In summary, while SPEM introduces slightly higher computational cost than the likelihood-only, typicality, complexity, and GMM methods due to feature extraction, it remains much more efficient than the likelihood ratio and LID methods.

