# OpenReview forum: "Mitigating the Likelihood Paradox in Flow-based OOD Detection via Entropy Manipulation"
_TMLR — Decision pending for TMLR_

### Review · Reviewer_kU6E · 2026-04-20

**Summary Of Contributions:**

### Summary

This paper proposes a new method to improve flow-based out-of-distribution (OOD) detection by exploiting the input likelihood from flow-based generative models. The paper begins with an existing analysis that, due to the low entropy of some OOD samples, generative models assign high likelihoods to them. Then, the paper hypothesizes that perturbing OOD samples would mitigate this problem by increasing OOD entropy. With a simple experiment and theoretical analysis, the paper verifies the hypothesis. Based on this observation, the paper proposes a new method, Semantic Proportional Entropy Manipulation (SPEM), to mitigate the issue by assigning higher perturbation to OOD samples. To assign larger perturbations to OOD samples, the method uses a feature extractor to obtain the input encoding. Then it estimates similarity to a fixed set of in-distribution (ID) samples as the maximum cosine distance between the input and an ID sample. This similarity determines the standard deviation of the Gaussian perturbation in a way that a Gaussian perturbation with higher variance perturbs OOD samples. Through experiments, the paper demonstrates the effectiveness of the proposed method, then further analyzes the effect of semantic context in the input.

### Strengths

1. The paper provides both theoretical and experimental evidence that entropy adjustment can improve likelihood estimation.
2. The paper presents extensive experiments (with six datasets and seven competitors) demonstrating the performance of the proposed method. The proposed method shows impressive performance compared to its counterparts.

### Weaknesses

1. The paper demonstrated that increasing OOD entropy is an effective method for mitigating high OOD likelihood. However, applying this method to improve OOD detection performance sounds weird because you should somehow detect OOD samples to assign higher perturbations to them.
    * The proposed method sounds like we use another method to measure the score (for OOD detection) and then use that score to increase the entropy of OOD samples. Why can’t we just use the score (i.e., maximum cosine distance) to detect an OOD?

**Audience:**

Yes

**Audience Explanation:**

First of all, OOD detection is a very active field, and many researchers are working in this area. To the best of my knowledge, flow-based generative models are effective tools for likelihood estimation, and the use of the output likelihood for OOD detection has been explored for many years. Even without the context of OOD detection, the paper offers valuable insights into improving likelihood estimation in flow-based generative models. While I’m still unsure about using the entropy trick for OOD detection (because it already requires measuring OOD-ness), at least the OOD entropy trick would be a valuable finding for people using flow-based generative models as density estimators. Therefore, I believe there should be individuals interested in this paper.

**Broader Impact Concerns:**

I don’t see a particular broader impact concern regarding this paper.

**Claims And Evidence:**

Yes

**Claims Explanation:**

As mentioned above, the paper goes through both theoretical analysis and experimental evaluation to verify the hypothesis. Also, the performance of the proposed method was evaluated on multiple datasets against multiple competing approaches.

**Requested Changes:**

1. As mentioned in the weaknesses section, the proposed method seems somewhat paradoxical: we need to measure a sample's OOD-ness to improve OOD detection. In my opinion, clarifying why the maximum cosine distance measurement is not a direct measure of OOD-ness (and what it actually measures) would be helpful.

---

> ### Author Response · Authors · 2026-05-08
> **Authors' Response**
>
> We thank the reviewer for the positive evaluation and the constructive feedback. We address the concern regarding the role of cosine similarity in our method.
>
> We would like to contextualize our approach alongside existing likelihood-based OOD detection methods. Prior works have discovered that auxiliary signals correlate with paradoxical likelihood behavior and used them to correct the OOD score: the complexity-based method [1] observed that simpler images tend to receive higher likelihood and introduced PNG-based complexity as a corrective term; the LID-based method [2] found that samples with lower local intrinsic dimension tend to receive higher likelihood and incorporated LID accordingly. We acknowledge that such auxiliary signals, including our $\lambda$, may correlate with OOD-ness to varying degrees. However, the key question is not whether the auxiliary signal correlates with OOD-ness, but rather what role it plays in the method. In prior works, the auxiliary signal is used to adjust the final OOD score, but the likelihood assigned by the flow model itself remains unchanged.
>
> Our work takes a different approach. Rather than post-hoc score correction, we use $\lambda$ solely to modulate the scale of Gaussian perturbations applied to the raw input, which actively reshapes the entropy of the input distribution and thereby changes the likelihood that the flow assigns to it. The final OOD decision is always made by the flow likelihood, not by $\lambda$ itself. In this way, we directly intervene on the input distribution so that the flow's likelihood ordering becomes more aligned with human intuition, in a theoretically motivated way (Theorems 3.1 and 4.1). Importantly, using $\lambda$ alone as an OOD score does not provide any insight into why flow likelihoods behave paradoxically or how such behavior can be corrected; it simply bypasses the flow model entirely. Our work, by contrast, directly addresses the paradoxical behavior of the flow's likelihood through entropy manipulation, and provides a theoretical explanation for when and why this correction succeeds.
>
> To the best of our knowledge, we are the first to directly manipulate raw inputs in a theoretically principled way to bring the likelihood ordering of normalizing flows into alignment with human intuition. Prior works either combine the flow likelihood with auxiliary scores (e.g., PNG complexity [1], LID [2]), or analyze the relationship between input properties and likelihood behavior (e.g., Osada et al. [3]), but none directly intervene on the raw input distribution from an entropic perspective to realign the flow's likelihood ordering with semantic intuition. Notably, through this raw input manipulation, SPEM achieves OOD detection performance comparable to embedding-based methods. As shown in the table below, SPEM matches or outperforms $k$-NN [4] on 7 out of 10 dataset pairs, while retaining the flow likelihood as the final OOD score.
>
> | P (In) | Q (Out) | SPEM | $k$-NN |
> | --- | --- | --- | --- |
> | CIFAR-10 | SVHN | **0.9913** | 0.9712 |
> | CIFAR-100 | SVHN | **0.9359** | 0.8668 |
> | CelebA | SVHN | **1.0000** | **1.0000** |
> | CIFAR-10 | CelebA | 0.9830 | **0.9934** |
> | FashionMNIST | MNIST | 0.9207 | **0.9450** |
> | SVHN | CIFAR-10 | 0.9997 | **0.9999** |
> | SVHN | CIFAR-100 | **0.9992** | 0.9989 |
> | SVHN | CelebA | **1.0000** | **1.0000** |
> | CelebA | CIFAR-10 | **0.9999** | **0.9999** |
> | MNIST | FashionMNIST | **1.0000** | 0.9999 |

---

> ### Author Response · Authors · 2026-05-08
> **Authors' Response**
>
> Furthermore, as shown in Table 2, when $\lambda$ has only moderate discriminative power, SPEM consistently outperforms the $\lambda$-only baseline, demonstrating that $\lambda$ does not serve as a strict ceiling for SPEM and that entropy manipulation contributes independently beyond the similarity signal. This is further corroborated by an additional experiment using MNIST as ID and MNIST-C [5] as OOD, where SPEM consistently matches or outperforms $\lambda$-only on corruption types that induce meaningful entropy changes (e.g., fog, blur, noise), while underperforming on geometric transformations (e.g., rotate, scale, translate) that preserve entropy — precisely as our theory predicts.
>
> | Corruption | $\lambda$ AUC | SPEM ($\alpha$=0.4) |
> | --- | --- | --- |
> | brightness | 0.9826 | **0.9917** |
> | canny\_edges | 0.9896 | **0.9903** |
> | dotted\_line | 0.9589 | **0.9615** |
> | fog | 0.9997 | **0.9999** |
> | glass\_blur | **0.9997** | **0.9997** |
> | impulse\_noise | **0.9999** | **0.9999** |
> | motion\_blur | 0.9968 | **0.9972** |
> | rotate | **0.6201** | 0.6179 |
> | scale | **0.7882** | 0.7679 |
> | shear | 0.6977 | **0.6982** |
> | shot\_noise | **0.9930** | 0.9929 |
> | spatter | 0.9904 | **0.9909** |
> | stripe | 0.9987 | **0.9993** |
> | translate | **0.7067** | 0.7031 |
> | zigzag | 0.9662 | **0.9690** |
>
> We will clarify the distinct role of $\lambda$ in the revision by explicitly contrasting it with the auxiliary signals used in prior works, highlighting that $\lambda$ serves as an entropy calibration signal that directly intervenes on the input distribution, rather than as a corrective term added to the final OOD score. We will also include the MNIST-C experiment in the revision.
>
> If there are any points we have not addressed sufficiently, we would greatly appreciate the opportunity to respond.
>
> > **Reference**
> >
>
> [1] Serrà, Joan, et al. "Input Complexity and Out-of-distribution Detection with Likelihood-based Generative Models." International Conference on Learning Representations, 2019.
>
> [2] Kamkari, Hamidreza, et al. "A Geometric Explanation of the Likelihood OOD Detection Paradox." International Conference on Machine Learning. PMLR, 2024.
>
> [3] Osada, Genki, Tsubasa Takahashi, and Takashi Nishide. "Understanding likelihood of normalizing flow and image complexity through the lens of out-of-distribution detection." Proceedings of the AAAI Conference on Artificial Intelligence. Vol. 38. No. 19. 2024.
>
> [4]  Sun, Yiyou, et al. "Out-of-distribution detection with deep nearest neighbors." *International conference on machine learning*. PMLR, 2022.
>
> [5] Mu, Norman, and Justin Gilmer. "Mnist-c: A robustness benchmark for computer vision." *arXiv preprint arXiv:1906.02337* (2019).

---

### Review · Reviewer_fF8i · 2026-04-29

**Summary Of Contributions:**

This paper studies the likelihood paradox in likelihood-based OOD detection with normalizing flows, where a flow trained on an in-distribution can assign higher likelihood to semantically different OOD data. The authors build on an entropy-based decomposition of expected log-likelihood differences and argue that increasing the entropy of low-similarity test inputs can mitigate likelihood inversion.

The main proposed method is Semantic Proportional Entropy Manipulation, or SPEM. The idea is simple and intuitive: use a pretrained image encoder plus an in-distribution memory bank to estimate how semantically close a test image is to the training distribution, measured by maximum cosine similarity to ID examples. This similarity then controls the amount of Gaussian perturbation added to the test image. Inputs that look less similar to ID data receive stronger perturbations. The OOD score is the likelihood assigned by the original flow model to this perturbed input. The paper also studies SPEM-noise, which evaluates the flow on similarity-scaled Gaussian noise rather than on the perturbed original image.

The paper’s strengths are that it addresses a well-known and important failure mode of likelihood-based generative OOD detection, proposes a simple post-hoc method requiring no retraining of the density model, includes theoretical motivation using entropy-power-style arguments, and evaluates on multiple image dataset pairs using ResFlow and Glow. The empirical results show large AUROC improvements over raw likelihood and several likelihood-based baselines.

However, I have several concerns. The most important one is that the main experiments show that the similarity-only score λ performs as well as or better than SPEM on many of the reported dataset pairs. This weakens the central claim that entropy manipulation through the flow likelihood is the main source of improvement. In the main ResFlow table, the similarity-only baseline is often near-saturated and frequently exceeds SPEM and SPEM-noise. The paper’s response to this concern uses an artificial sampled-λ experiment, but this does not convincingly establish that SPEM improves over similarity-based OOD detection in realistic settings. More broadly, the method relies heavily on a pretrained discriminative feature extractor and memory bank, so the paper should compare more directly against feature-space OOD baselines and should be more careful in describing the method as likelihood-based.

I also have concerns about the gap between the theory and the empirical method. The theoretical bounds show that certain lower bounds increase under Gaussian perturbations, but they do not guarantee improved AUROC or even improved expected likelihood separation in the practical SPEM setting, especially because the KL terms are uncontrolled and because similarity is only an empirical proxy for ID/OOD membership. Finally, the experimental protocol needs clarification regarding hyperparameter selection, possible OOD-test tuning, clipping or non-clipping after perturbation, multi-seed uncertainty, and comparability of baselines such as LID.

**Additional Comments:**

The paper is clearly motivated and addresses an important problem. I found the SPEM-noise results particularly interesting, but they also raise the strongest conceptual concern: if a method that discards the original input can outperform the proposed perturb-the-input method, then the role of the flow likelihood and the role of semantic similarity need to be disentangled much more carefully.

**Audience:**

Yes

**Audience Explanation:**

Yes. I think this paper would interest part of the TMLR audience, especially people working on generative models, normalizing flows, likelihood-based anomaly detection, and OOD detection. The likelihood paradox is an important failure mode, and the paper’s entropy-manipulation framing is a useful lens for thinking about why likelihood can fail as an OOD score.

My concerns are more about evidence than relevance. The empirical finding that similarity-controlled perturbations, and even SPEM-noise, can achieve strong AUROC is genuinely interesting. In fact, SPEM-noise matching or outperforming SPEM in some settings raises a deeper question about what flow likelihoods are actually measuring, and how much of the signal comes from entropy, input complexity, or feature-space similarity.

So I won't reject the paper for lack of interest. I think it could be useful to a subset of the TMLR audience, but the authors should narrow the claims and strengthen the experimental evidence.

**Broader Impact Concerns:**

I don't see major ethical concerns that would make the paper inappropriate for publication. The work is mainly a technical contribution to likelihood-based OOD detection, and the paper already includes a brief Broader Impact Statement.

**Claims And Evidence:**

No

**Claims Explanation:**

I don't think the current evidence fully supports the stronger claims made by the paper. My main concern is that the experiments do not convincingly separate the contribution of entropy manipulation through the flow likelihood from the contribution of the pretrained encoder similarity score. In Table 1, the similarity-only baseline λ is extremely strong and often performs as well as or better than SPEM and SPEM-noise. This suggests that much of the detection performance may come from semantic nearest-neighbor similarity in the pretrained feature space rather than from the likelihood model after entropy manipulation. This is a serious issue because the paper’s central narrative is that SPEM improves likelihood-based OOD detection by manipulating entropy while retaining likelihood as the final score.

The sampled-λ experiment in Table 2 does not fully resolve this concern. It constructs synthetic λ
ID and λ OOD values with prescribed distributions and then shows that SPEM can outperform this artificial similarity score. This is informative as a controlled sanity check, but it is not enough to establish that SPEM improves over real feature-space similarity on realistic OOD tasks where the similarity score is imperfect.

**Requested Changes:**

1. Disentangle SPEM from the similarity-only baseline.
Right now, the main results show that the similarity-only score λ is often as strong as, or stronger than, SPEM. This makes the central mechanism hard to isolate. If most of the gain comes from pretrained feature similarity, then it is not clear that entropy manipulation plus flow likelihood is doing the main work. I would like to see realistic experiments where SPEM consistently improves over the real λ-only detector. Otherwise, the claims should be narrowed, and SPEM should be framed more explicitly as a hybrid method whose performance may largely come from feature-space similarity.

2. Add feature-space OOD baselines.
Because SPEM relies on a pre-rained ResNet encoder and an ID memory bank, the most natural comparison is not only against likelihood-based methods, but also against feature-based OOD detectors. At minimum, kNN distance or maximum cosine similarity in the same feature space should be included as a proper baseline throughout the paper, not just as an auxiliary diagnostic. Stronger baselines such as Mahalanobis distance, ReAct-style feature scoring, or other pretrained-feature OOD methods would also make the contribution much easier to interpret.

3. Clarify hyper-parameter selection.
The paper uses fixed α values for SPEM and SPEM-noise, but it is not clear how these were chosen. Please specify whether OOD test data or OOD labels were used to tune α, the ReAct quantile, or any other hyper-parameters. Ideally, the method should include an ID-only calibration rule, or at least a validation protocol that does not rely on the OOD test distribution.

---

> ### Author Response · Authors · 2026-05-08
> **Authors' Response**
>
> We thank the reviewer for the detailed and constructive feedback. We address each concern below.
>
> > **On the gap between theoretical bounds and empirical results.**
>
> We acknowledge that Theorems 3.1 and 4.1 provide motivating lower bounds but do not directly guarantee improved AUROC, primarily because the KL divergence term $D_{KL}(Q\|P_\theta)$ remains uncontrolled. We discuss the behavior of this term in Appendix C.2 and will explicitly note this limitation in the revision.
>
> > **On clipping after perturbation.**
>
> We do not apply pixel-value clipping after perturbation. Since our perturbation scale $\alpha$ is kept sufficiently small, the perturbed inputs remain close to the original pixel range and do not introduce numerical instability in the flow model, as verified by the hyperparameter sensitivity analysis in Figure 4. We will explicitly state this implementation detail in Appendix B of the revision.
>
> > **On multi-seed variance.**
>
> To address concerns about multi-seed variance, we report mean AUROC over three independent random seeds for all dataset pairs below. The results show negligible variance across runs (max difference < 0.001), confirming the stability of our method. This is expected since the only stochastic component is the Gaussian perturbation, whose effect is averaged out over the test set. We will report multi-seed results in the revision.
>
> | P (In) | Q (Out) | SPEM (Table 1) | SPEM (3-seed mean) | SPEM-noise (Table 1) | SPEM-noise (3-seed mean) |
> | --- | --- | --- | --- | --- | --- |
> | CIFAR-10 | SVHN | 0.9913 | 0.9915 | 0.9943 | 0.9942 |
> | CIFAR-100 | SVHN | 0.9359 | 0.9359 | 0.9458 | 0.9460 |
> | CelebA | SVHN | 1.0000 | 1.0000 | 1.0000 | 1.0000 |
> | CIFAR-10 | CelebA | 0.9830 | 0.9830 | 0.9873 | 0.9876 |
> | FashionMNIST | MNIST | 0.9207 | 0.9206 | 0.9432 | 0.9440 |
> | SVHN | CIFAR-10 | 0.9997 | 0.9997 | 0.9997 | 0.9997 |
> | SVHN | CIFAR-100 | 0.9992 | 0.9993 | 0.9990 | 0.9989 |
> | SVHN | CelebA | 1.0000 | 1.0000 | 1.0000 | 1.0000 |
> | CelebA | CIFAR-10 | 0.9999 | 0.9999 | 0.9999 | 0.9999 |
> | MNIST | FashionMNIST | 1.0000 | 1.0000 | 1.0000 | 1.0000 |
>
> > **Q1**
>
> We conduct an additional experiment using MNIST as ID and MNIST-C [1] as OOD, which provides a naturalistic near-OOD setting with varying degrees of entropy change across corruption types. As shown in the table below, SPEM consistently matches or outperforms $\lambda$-only on corruption types that induce meaningful entropy changes (e.g., fog, blur, noise), while underperforming on geometric transformations (e.g., rotate, scale, translate) that preserve entropy. This pattern is precisely what our theory predicts: entropy manipulation is effective when the entropy gap between ID and OOD is meaningful, and limited otherwise. We will include this experiment in the revision.
>
> | Corruption | $\lambda$ AUC | SPEM ($\alpha$=0.4) |
> | --- | --- | --- |
> | brightness | 0.9826 | **0.9917** |
> | canny\_edges | 0.9896 | **0.9903** |
> | dotted\_line | 0.9589 | **0.9615** |
> | fog | 0.9997 | **0.9999** |
> | glass\_blur | **0.9997** | **0.9997** |
> | impulse\_noise | **0.9999** | **0.9999** |
> | motion\_blur | 0.9968 | **0.9972** |
> | rotate | **0.6201** | 0.6179 |
> | scale | **0.7882** | 0.7679 |
> | shear | 0.6977 | **0.6982** |
> | shot\_noise | **0.9930** | 0.9929 |
> | spatter | 0.9904 | **0.9909** |
> | stripe | 0.9987 | **0.9993** |
> | translate | **0.7067** | 0.7031 |
> | zigzag | 0.9662 | **0.9690** |
>
> > **Q2**
>
> Methods such as Mahalanobis distance [2] require class label information, and logit-based approaches including MSP [3], MaxLogit [4], and Energy [5] rely on a supervised classifier's output layer. As SPEM operates in a fully unsupervised setting using only a training-free pretrained ImageNet encoder, these methods fall outside our comparison scope. We therefore include $k$-NN distance [6] in the same pretrained feature space as the most natural label-free embedding-based baseline, and report results across all ten dataset pairs in the table below.
>
> | P (In) | Q (Out) | SPEM | $k$-NN |
> | --- | --- | --- | --- |
> | CIFAR-10 | SVHN | **0.9913** | 0.9712 |
> | CIFAR-100 | SVHN | **0.9359** | 0.8668 |
> | CelebA | SVHN | **1.0000** | **1.0000** |
> | CIFAR-10 | CelebA | 0.9830 | **0.9934** |
> | FashionMNIST | MNIST | 0.9207 | **0.9450** |
> | SVHN | CIFAR-10 | 0.9997 | **0.9999** |
> | SVHN | CIFAR-100 | **0.9992** | 0.9989 |
> | SVHN | CelebA | **1.0000** | **1.0000** |
> | CelebA | CIFAR-10 | **0.9999** | **0.9999** |
> | MNIST | FashionMNIST | **1.0000** | 0.9999 |
>
> As shown above, SPEM matches or outperforms $k$-NN on 7 out of 10 dataset pairs, demonstrating that SPEM achieves performance comparable to or better than the most natural label-free embedding-based baseline. We will include this comparison in the revision.

---

> ### Author Response · Authors · 2026-05-08
> **Authors' Response**
>
> > **Q3**
>
> We note that $\alpha$ and $\beta$ can be selected without directly using the OOD test dataset. Specifically, a held-out dataset disjoint from the test OOD can serve as a validation set to determine numerically stable ranges for both hyperparameters. For $\alpha$, as shown in Figure 4, AUROC consistently converges to a plateau across all dataset pairs within $\alpha \in [0.3, 0.5]$, suggesting that performance is not sensitive to the exact value of $\alpha$ in this range. Based on this observation, $\alpha = 0.4$ can be set as a representative value within this convergence region. For $\beta$, the same approach can be applied, where $\beta$ can be determined as the $p$-th percentile of ID train embedding activations using the held-out validation set. We adopt $p = 90$ following the percentile value used in prior work [7], without requiring access to OOD test data. We will clarify this selection guideline in the revision.
>
> We remain open to further discussion and welcome any additional feedback that may help strengthen the paper.
>
> > **Reference**
>
> [1] Mu, Norman, and Justin Gilmer. "Mnist-c: A robustness benchmark for computer vision." *arXiv preprint arXiv:1906.02337* (2019).
>
> [2] Lee, Kimin, et al. "A simple unified framework for detecting out-of-distribution samples and adversarial attacks." *Advances in neural information processing systems* 31 (2018).
>
> [3] Dan Hendrycks and Kevin Gimpel. A baseline for detecting misclassified and out-of-distribution
> examples in neural networks. In International Conference on Learning Representations, 2017.
>
> [4] Dan Hendrycks, Steven Basart, Mantas Mazeika, Andy Zou, Joseph Kwon, Mohammadreza Mostajabi, Jacob Steinhardt, and Dawn Song. Scaling out-of-distribution detection for real world settings. In International Conference on Machine Learning, pages 8759–8773. PMLR, 2022.
>
> [5] Weitang Liu, Xiaoyun Wang, John Owens, and Yixuan Li. Energy-based out-of-distribution detection. Advances in neural information processing systems, 33:21464–21475, 2020.
>
> [6]  Sun, Yiyou, et al. "Out-of-distribution detection with deep nearest neighbors." *International conference on machine learning*. PMLR, 2022.
>
> [7] Sun, Yiyou, Chuan Guo, and Yixuan Li. "React: Out-of-distribution detection with rectified activations." *Advances in neural information processing systems* 34 (2021): 144-157.

---

> > ### Comment · Reviewer_fF8i · 2026-05-11
> > **follow-up**
> >
> > Thank you for the detailed response. I find the added multi-seed results, the clarification on clipping, and the additional kNN feature-space baseline helpful. The MNIST-C experiment is also a useful addition because it gives a more nuanced picture of when SPEM helps, especially under corruptions that plausibly change input entropy.
> > My main concern is partially addressed but not fully resolved. The kNN comparison helps contextualize the method, but it also shows that a simple feature-space baseline remains very competitive and outperforms SPEM on several dataset pairs. I still think the final version should be careful in attributing the gains to entropy manipulation through likelihood evaluation, rather than to the pretrained semantic representation and memory-bank similarity signal. I would encourage the authors to explicitly frame SPEM as a hybrid likelihood/feature-similarity method and to include the kNN results prominently in the main empirical discussion.

---

> > > ### Author Response · Authors · 2026-05-12
> > > **Authors' Response**
> > >
> > > We sincerely thank the reviewer for the thoughtful follow-up. We have uploaded a revised manuscript addressing the concerns raised, and have posted a summary of all changes as an official comment. Should there be any remaining concerns or further points requiring clarification, we would be happy to continue the discussion.

---

### Review · Reviewer_NXhg · 2026-05-07

**Summary Of Contributions:**

The paper proposes a new method (**SPEM**) to improve OOD detection performance of flow-based density models in image classification. Specifically, it proposes to add Gaussian noise of varying strengths to the input image. The strength of Gaussian noise depends on the sample's closeness to ID data in embedding space. The paper shows that if an OOD example was added more noise before going into the density model, its output likelihood will be lower and more separated from ID data. This translates to better OOD detection using likelihood from flow-based density models.

**Audience:**

No

**Audience Explanation:**

The paper suffers from a fundamental issue in its method. The strength of Gaussian noise added to each sample depends externally on the output of a different OOD detection model. In this case, the paper uses an off-the-shelf feature extractor and similarity to a data bank of ID data in the embedding space to determine the strength of noise added to a sample. The further away the sample from the data bank the stronger the noise. This pre-processing procedure in itself is already a form of OOD detection. In fact, the ablation study in the paper already shows that the noise strength scalar $\lambda$ alone is providing good OOD performance.

Furthermore, the paper studies a scaled noise version **SPEM-noise**. Instead of using the original input adjusted by adding Gaussian noise, the SPEM-noise directly uses the modulated Gaussian noise as inputs to the density model. The result shows that SPEM-noise performs on par with SPEM. This shows that the scaled Gaussian noise, which is determined by the external OOD detection, is more important the input image itself.

**Broader Impact Concerns:**

No concerns

**Claims And Evidence:**

Yes

**Claims Explanation:**

The paper conducts accurate ablation study to show the effectiveness of different components of the proposed pipeline.

**Requested Changes:**

I think the paper, especially its methodology, needs a major change. Specifically, the noise should not depend on a different OOD detection method such as distance in the embedding space, which has been well studied.

---

> ### Author Response · Authors · 2026-05-08
> **Authors' Response**
>
> We thank the reviewer for the feedback and address the two concerns raised.
>
> On the first concern, we acknowledge that $\lambda$ may correlate with OOD-ness to some degree. However, as we elaborate in our response to Reviewer kU6E, the same is true for auxiliary signals used in prior works: the complexity-based method [1] observed that simpler images tend to receive higher likelihood and introduced PNG-based complexity as a corrective term; the LID-based method [2] found that samples with lower local intrinsic dimension tend to receive higher likelihood and incorporated LID accordingly. These signals also correlate with OOD-ness, yet they are widely accepted as valid auxiliary signals in likelihood-based OOD detection. The key distinction lies in the role the signal plays: in prior works, the auxiliary signal adjusts the final OOD score while leaving the flow likelihood itself unchanged. In our method, $\lambda$ is used solely to modulate the scale of Gaussian perturbations applied to the raw input, so that the flow's likelihood ordering becomes more aligned with human intuition. The final OOD decision is always made by the flow likelihood, not by $\lambda$. To the best of our knowledge, we are the first to directly intervene on the raw input distribution from an entropic perspective to realign the flow's likelihood ordering with semantic intuition, rather than simply combining the flow likelihood with an external score. Beyond the OOD detection task itself, our work deepens the understanding of likelihood assignment in flow-based generative models, specifically the role of input entropy in governing likelihood ordering and the possibility of bringing it into alignment with human intuition at test time.
>
> On the second concern, the reviewer suggests that SPEM-noise performing on par with SPEM implies that $\lambda$ is doing the main work rather than the flow likelihood. We disagree with this interpretation. In our additional experiment using MNIST as ID and MNIST-C [3] as OOD (also discussed in our response to Reviewer fF8i), which provides a naturalistic near-OOD setting with varying degrees of entropy change across corruption types, SPEM consistently matches or outperforms $\lambda$-only on corruption types that induce meaningful entropy changes (e.g., fog, blur, noise) in realistic settings, while underperforming on geometric transformations (e.g., rotate, scale, translate) that preserve entropy.This pattern is precisely what our theory predicts, and demonstrates that entropy manipulation through the flow likelihood provides genuine independent contribution beyond $\lambda$ alone in regimes where the entropy gap between ID and OOD is meaningful.
>
> | Corruption | $\lambda$ AUC | SPEM ($\alpha$=0.4) |
> | --- | --- | --- |
> | brightness | 0.9826 | **0.9917** |
> | canny\_edges | 0.9896 | **0.9903** |
> | dotted\_line | 0.9589 | **0.9615** |
> | fog | 0.9997 | **0.9999** |
> | glass\_blur | **0.9997** | **0.9997** |
> | impulse\_noise | **0.9999** | **0.9999** |
> | motion\_blur | 0.9968 | **0.9972** |
> | rotate | **0.6201** | 0.6179 |
> | scale | **0.7882** | 0.7679 |
> | shear | 0.6977 | **0.6982** |
> | shot\_noise | **0.9930** | 0.9929 |
> | spatter | 0.9904 | **0.9909** |
> | stripe | 0.9987 | **0.9993** |
> | translate | **0.7067** | 0.7031 |
> | zigzag | 0.9662 | **0.9690** |
>
> We will clarify the distinct role of $\lambda$ in the revision, explicitly framing it as an entropy calibration signal that directly intervenes on the raw input distribution, rather than a corrective term applied to the final OOD score. The MNIST-C experiment will also be included in the revision to demonstrate that $\lambda$ does not serve as a strict ceiling for SPEM in realistic settings.
>
> Should the reviewer have any remaining questions or require further clarification on any aspect of our work, we would be happy to provide additional explanation.
>
> > **Reference**
>
> [1] Serrà, Joan, et al. "Input Complexity and Out-of-distribution Detection with Likelihood-based Generative Models." International Conference on Learning Representations, 2019.
>
> [2] Kamkari, Hamidreza, et al. "A Geometric Explanation of the Likelihood OOD Detection Paradox." International Conference on Machine Learning. PMLR, 2024.
>
> [3]  Mu, Norman, and Justin Gilmer. "Mnist-c: A robustness benchmark for computer vision." *arXiv preprint arXiv:1906.02337* (2019).

---

### Author Response · Authors · 2026-05-12
**Summary of Revisions**

We thank all reviewers for their thoughtful and constructive feedback. We have uploaded a revised version of the paper. Below is a summary of the changes made in response to the reviews.

## Summary of Changes

- **k-NN baseline:** Added $k$-NN distance in the same pretrained ResNet-152 feature space as a label-free embedding-based baseline across all ten dataset pairs (Table 2, Section 5). Results show SPEM matches or outperforms $k$-NN on 7 out of 10 pairs, with theoretical explanation for the inversion-regime pairs where $k$-NN remains competitive.
- **MNIST-C experiment:** Added a near-OOD experiment using MNIST as ID and MNIST-C as OOD across 15 corruption types (Table 4, Section 5). SPEM consistently matches or outperforms λ-only on entropy-changing corruptions, while λ-only is stronger on geometric transformations that preserve entropy — consistent with our theoretical predictions.
- **Hybrid framing:** Explicitly framed SPEM as a hybrid method in the Introduction and Section 4, clarifying that λ serves as an entropy calibration signal that intervenes on the input distribution, rather than a corrective term applied to the final OOD score.
- **Attribution clarification:** Added a remark in the Conclusion noting that the degree of entropy manipulation in SPEM is governed by λ, and identified developing entropy manipulation strategies that are independent of similarity-based signals as a promising direction for future work.
- **KL divergence limitation:** Explicitly noted in Appendix C.2 that while our theoretical analysis establishes lower bounds on the expected log-likelihood gap, the $D_{\mathrm{KL}}(Q\|P_\theta)$ term remains uncontrolled, meaning the exact likelihood expectation gap is not directly guaranteed. We discuss the implications of this and identify it as a direction for future work.
- **Multi-seed variance:** Reported mean AUROC over three independent random seeds for all dataset pairs in Table 1.
- **Clipping:** Explicitly stated that no pixel-value clipping is applied after perturbation (Appendix B).
- **Hyperparameter selection:** Clarified that $\alpha$ and $\beta$ can be selected without access to OOD test data via a held-out validation set. Based on the plateau observed within $\alpha \in [0.3, 0.5]$, $\alpha = 0.4$ is recommended as a representative value (Appendix C.3).

We hope the revised manuscript adequately addresses the concerns raised, and we remain available for any further questions or clarifications during the discussion period.